# Near, far: Patch-ordering enhances vision foundation models' scene understanding

**Valentinos Pariza[1]\*, Mohammadreza Salehi[1]\*,**
**Gertjan Burghouts[2], Francesco Locatello[3], Yuki M. Asano[1]**
[1] University of Amsterdam, [2] TNO, [3] Institute of Science and Technology Austria, \* Equal contribution
valentinos.pariza@uva.nl, s.salehidehnavi@uva.nl

## Abstract

We introduce `NeCo`: Patch Neighbor Consistency, a novel self-supervised training loss that enforces patch-level nearest neighbor consistency across a student and teacher model. Compared to contrastive approaches that only yield binary learning signals, *i.e.* 'attract' and 'repel', this approach benefits from the more fine-grained learning signal of sorting spatially dense features relative to reference patches. Our method leverages differentiable sorting applied on top of pretrained representations, such as DINOv2-registers to bootstrap the learning signal and further improve upon them. This dense post-pretraining leads to superior performance across various models and datasets, despite requiring only 19 hours on a single GPU. This method generates high-quality dense feature encoders and establishes several new state-of-the-art results such as +2.3 % and +4.2% for non-parametric in-context semantic segmentation on ADE20k and Pascal VOC, +1.6% and +4.8% for linear segmentation evaluations on COCO-Things and -Stuff and improvements in the 3D understanding of multi-view consistency on SPair-71k, by more than 1.5%.

## 1 Introduction

Dense self-supervised learning trains feature extractors to produce representations for every pixel or patch of an image without supervision. This field has seen substantial advancements in recent years, notably improving unsupervised semantic segmentation (Ziegler & Asano, 2022; Salehi et al., 2023; Araslanov et al., 2021; Stegmüller et al., 2023; Wang et al., 2021), object-centric representation learning (Zadaianchuk et al., 2023), and other dense downstream tasks like object tracking and object detection (Hénaff et al., 2021; 2022; Lebailly et al., 2023; Salehi et al., 2023).

One particularly interesting use-case of densely pretrained encoders was developed by Balazevic et al. (2023). They propose to solve semantic segmentation by posing it as a nearest-neighbor retrieval problem utilizing the features of the spatial patches. This non-parametric method not only mirrors in-context learning in large language models (LLMs) (Brown et al., 2020) but also delivers rapid and robust performance, especially with limited data.

Building on this idea, we propose an inverted approach: using nearest-neighbor retrieval not just for evaluation but as a *training* mechanism for encoders. This approach promises a more fine-grained learning signal, enabling the capture of intricate visual details. For instance, the features of a tire should be closely related to one another, as well as to those of a car body, while remaining distinct from features of an airplane.

Unlike contrastive losses, which offer a binary 'attract' or 'repel' signal, this provides a much richer, continuous learning signal. Additionally, it avoids the pitfalls of reconstruction-based methods like MAE (He et al., 2022), where low-level RGB patches that are visually similar may not carry similar semantics. By operating strictly in the deep feature space, learning is guided by higher-level semantics rather than superficial pixel-level similarities. As a result, this method promises to yield models with deeply semantic spatial features specifically tailored for in-context tasks, enhancing their adaptability and robustness. However, this approach, while promising, presents two main challenges.

The first is the source of supervision. In the case of evaluation, ground-truth labels are used, yet we are interested in obtaining better *self*-supervised representations. While previous works (Balazevic

et al., 2023; Lebailly et al., 2023) address this by essentially converting dense learning to image-level learning via learnable pooling of patches, we offer a more practical and versatile solution. We simply start from already image-level pretrained models and adapt them further. We term this stage *dense post-pretraining* and demonstrate that it is an effective and fast solution to this problem, taking only 19 hours on a single GPU for tuning for a ViT-S/16 model.

The second challenge is the discrete nature of nearest-neighbor retrieval, which does not yield gradients. To overcome this, we apply a differentiable sorting method proposed by Petersen et al. (2021), originally developed for ranking supervision, that we can use to backpropagate gradients. As we demonstrate empirically, this results in a more efficient and effective algorithm.

Our method enforces Patch Neighbor Consistency, so we term it NeCo. We show that it can be applied on top of image-level pretrained models such as DINO (Caron et al., 2021), and densely trained ones like iBOT (Zhou et al., 2022), Leopart (Ziegler & Asano, 2022), CrIBO (Lebailly et al., 2023) and DINOv2 (Oquab et al., 2023; Darcet et al., 2024) to obtain superior features for in-context scene understanding. Despite not being as close to our NeCo training task, our method also consistently excels on downstream benchmarks such as 3D understanding, unsupervised semantic segmentation and also full-finetuning semantic segmentation, where it even improves upon the state-of-the-art DINOv2-R model (Darcet et al., 2024).

Overall, our contributions can be summarized as follows:

- We propose a new post-pretraining adaptation that applies a dense, patch-sorting-based self-supervised objective, NeCo, applicable to any pretrained Vision Transformer
- We demonstrate NeCo's utility by applying it to six different backbones and evaluating it on five datasets and five evaluation protocols, achieving performance gains from 6% to 16%.
- We set several new state-of-the-art performances, for example on the in-context segmentation benchmark of Balazevic et al. (2023), we outperform the previous methods such as CrIbo and DINOv2 on Pascal VOC and ADE20k by 4% to 13% across different metrics.

## 2 RELATED WORKS

**Dense Self-supervised Learning.** Dense self-supervised learning methods aim to generate categorizable representations at the pixel or patch level, rather than the image level. This field has gained significant attention (Salehi et al., 2023; Ziegler & Asano, 2022; Araslanov et al., 2021; Stegmüller et al., 2023; Lebailly et al., 2023; Balazevic et al., 2023; Hwang et al., 2019; Liu et al., 2020; Hénaff et al., 2022; Van Gansbeke et al., 2021; Hénaff et al., 2022; 2021; Yun et al., 2022) due to the observation that image-level self-supervised methods (Caron et al., 2018; Asano et al., 2019; Chen et al., 2020; Grill et al., 2020; Caron et al., 2021; Izacard et al., 2021; Oquab et al., 2023) do not necessarily produce expressive dense representations (Balazevic et al., 2023; Hénaff et al., 2022; He et al., 2019; Purushwalkam & Gupta, 2020).

CroC (Stegmüller et al., 2023) is a recent method that has proposed a dense self-supervised loss to address the issue. It applies joint clustering between different views, ensuring that cluster centers capturing the same object are similar. Leopart (Ziegler & Asano, 2022) improves dense representations by applying a dense clustering loss to pretrained models. Extending this concept, TimeTuning (Salehi et al., 2023) has demonstrated that finetuning pretrained backbones over the temporal dimension of unlabeled videos enhances dense reasoning capabilities. Recently, Hummingbird (Balazevic et al., 2023) has proposed a dense loss that leverages attention within and across images during training, showing strong in-context scene understanding during evaluation. CrIBo (Lebailly et al., 2023) takes this further by explicitly enforcing cross-image nearest neighbor consistency between image objects, achieving state-of-the-art results.

We similarly adopt nearest neighbor consistency due to its promising results in in-context scene understanding but with two major differences: (1) instead of using pooled versions of patches at either the image or object level, we directly apply it to *patch* features; (2) in addition to nearest neighbor consistency, we ensure that the *order* of neighbors for the same patches from different views is similar. These changes result in more semantic patch-level features, directly enhancing in-context scene understanding and stabilizing training, as there is no need to infer object-level features through clustering methods, which can be unstable during training.

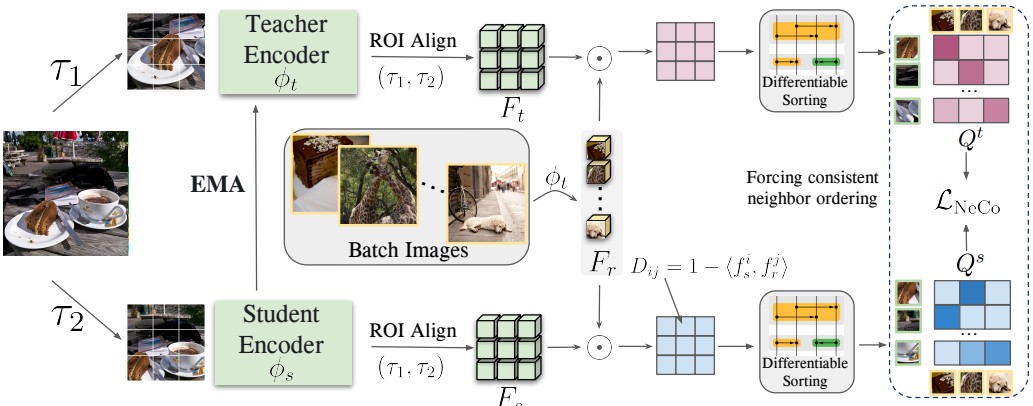

Figure 1: **NeCo overview.** Given an input image $I$, two augmentations $\tau_1$ and $\tau_2$ are applied to create two different views, which are processed by the teacher and student encoders, $\phi_t$ and $\phi_s$ respectively. The teacher encoder is updated using Exponential Moving Average (EMA). The encoded features are then aligned using ROI Align and compared with reference features $F_r$ obtained by applying $\phi_t$ to other batch images. Next, pairwise distances $D_{ij}$ between $F_s$ and $F_r$, as well as between $F_t$ and $F_r$, are computed using cosine similarity. These distances are then sorted using differentiable sorting and utilized to force nearest order consistency across the views through the NeCo loss.

**Unsupervised Object Segmentation.** Several works specifically target unsupervised object segmentation (Seitzer et al., 2022; Löwe et al., 2023; Bao et al., 2023; Siméoni et al., 2023; Zadaianchuk et al., 2023; Wang et al., 2023; Hamilton et al., 2022; Lan et al., 2023; Zadaianchuk et al., 2022). The goal of these works is not to learn semantic patch-level representations; instead, they often utilize the existing information in frozen pretrained backbones and train another model to explicitly solve semantic segmentation. For instance, Seitzer et al. (2022) train a slot-attention encoder and decoder module (Locatello et al., 2020) to reconstruct DINO (Caron et al., 2021) pretrained features with a few slots for each input image. This process enables the creation of per-image object cluster maps, where each slot represents a distinct object or part of an object within the image, based on the features it predicts.

In contrast, our approach learns distinct features for various objects and employs dense representations as intermediaries for dense tasks like semantic segmentation. These features can then be used to develop per-dataset cluster maps for semantic segmentation use cases.

## 3 PATCH NEIGHBOR CONSISTENCY

The goal is to develop a feature space in which, for a given input, patches representing the same object exhibit similar features, whereas patches representing different objects show distinct features. A key challenge for a self-supervised method in this process is defining the similarity between image patches. Although patches from the same object (*e.g.* a cat) are expected to be more similar to each other than to those from different objects (*e.g.* a dog), they may still depict different parts of the object (*e.g.* a cat's tail and legs).

Therefore, a learned dense feature space must provide an *ordering* of similarities to ensure that patches from the same object and its parts are correctly distinguished from those of other objects. To this end, our method works by extracting dense features of the inputs, finding their pair-wise distances, and forcing a consistency between the order of nearest neighbors within a batch across two views. While contrastive and clustering-based methods (Chen et al., 2020; Caron et al., 2020; Ziegler & Asano, 2022; Salehi et al., 2023) can be seen as potential substitutes for sorting in aligning positive views, they only enforce similarity without capturing the nuanced ordering constraints necessary for structured relationships. Moreover, they often rely on explicit negative samples, which limits their flexibility. In contrast, our sorting-based approach not only maintains relative similarities without the need for negatives but also prevents mode collapse by preserving subtle distinctions among patches, even within the same object. Figure 1 provides an overview of the method, which we describe in detail below.

**Feature Extraction and Alignment.** Given an input image $I$, two augmentations specified by parameters $\tau_1$ and $\tau_2$ are applied to create two different views $V_1$ and $V_2$. These views are then divided into $N = \lfloor \frac{H}{P} \rfloor \times \lfloor \frac{W}{P} \rfloor$ separate patches, where $H$ and $W$ represent the height and width of the input image, and $P$ represents the patch size. The patches are represented as $V_1^p = \left[ v_1^1, \ldots, v_N^1 \right]$ and $V_2^p = \left[ v_1^2, \ldots, v_N^2 \right]$, which are fed to the feature extractor.

We utilize the Vision Transformer (ViT) architecture (Dosovitskiy et al., 2020) as the backbone and employ a teacher-student framework, where the student and teacher models are denoted by $\phi_s$ and $\phi_t$, respectively. The teacher's weights are updated using the exponential moving average of the student's weights.

As the generated views cover different parts of the input, the extracted features do not necessarily correspond to the same objects. To address this, we align the features by applying ROI-Align (He et al., 2017), adjusted according to the crop augmentation parameters. This process creates spatially aligned dense features for the teacher and student networks, represented by $F_s \in \mathbb{R}^{N' \times d}$ and $F_t \in \mathbb{R}^{N' \times d}$. These features are then forced to maintain a consistent order of nearest neighbors, ensuring more robust and meaningful feature representations, as explained next.

**Pairwise Distance Computation.** To identify the nearest neighbors of the patches, it is necessary to extract features from other images in the batch and compute their distances with respect to $F_s$ and $F_t$. To achieve this, all batch images are fed through the teacher network, $\phi_t$, to obtain the batch features $F_B \in \mathbb{R}^{BN \times d}$. We sample a random fraction $f \ll 1$ of these patches to obtain the $R = fBN$ reference patches $F_r \in \mathbb{R}^{r \times d}$ which we use to compare the nearest neighbors of our $F_s$ and $F_t$ features. To this end, we compute distances based on cosine similarities,

$$D_s(i,j) = 1 - \frac{\langle F_s^i, F_r^j \rangle}{\|F_s^i\|\|F_r^j\|}, \tag{1}$$

$$D_t(i,j) = 1 - \frac{\langle F_t^i, F_r^j \rangle}{\|F_t^i\|\|F_r^j\|}, \tag{2}$$

$$i \in (1, \ldots, N'), \ \ j \in (1, \ldots, R), \tag{3}$$

Next, these distance matrices are sorted in a differentiable manner to produce a loss that enforces a similar sorting across the two views.

**Differentiable Sorting of Distances.** To determine the order of nearest neighbors from distance matrices, sorting is necessary. However, traditional sorting algorithms cannot propagate gradients because they use non-differentiable operations such as $d_i' \leftarrow \min(d_a, d_b)$ and $d_a' \leftarrow \max(d_a, d_b)$ to facilitate element swapping in the sequence for an ordering $a < b$. Given a sequence $s = (d_1, \ldots, d_R)$, where $R$ is the length of the sequence, We use relaxed, differentiable versions of these operations by defining their soft versions following recent work (Lee et al., 2017), as follows:

$$d_a' = \text{softmin}(d_a, d_b) := d_a f(d_b - d_a) + d_b f(d_a - d_b), \tag{4}$$

$$d_b' = \text{softmax}(d_a, d_b) := d_a f(d_a - d_b) + d_b f(d_b - d_a), \tag{5}$$

where the function $f(x) = \frac{1}{\pi} \arctan(\beta x) + 0.5$, and $\beta > 0$ is an inverse temperature parameter, specifying the steepness of the operator. This function is sigmoid-shaped and centered around $x = 0$. As $\beta$ approaches infinity, the relaxation converges to the discrete swap operation. This operation can be defined in an approximate permutation matrix $P_{\text{swap}}(d_i, d_j) \in \mathbb{R}^{L \times L}$, which is essentially an identity matrix except for the entries $P_{ii}$, $P_{ij}$, $P_{ji}$, and $P_{jj}$ defined as

$$P_{ii} = P_{jj} = f(d_j - d_i), \qquad P_{ij} = P_{ji} = f(d_i - d_j), \tag{6}$$

such that one step of swapping the pair $(d_i, d_j)$ in the sequence is equivalent to multiplying $P$ with that sequence. The final permutation matrix for the entire sequence is determined by the sorting algorithm employed. For example, in the odd-even sorting algorithm, the permutation matrix $P_t$ for a step $t$ is defined as:

$$P_t = \prod_{i \in M} P_{\text{swap}}(d_i, d_{i+1}), \tag{7}$$

where $M$ is the set of odd indices if $t$ is odd and the set of even indices if $t$ is even. The overall permutation matrix $Q$ is obtained by multiplying the permutation matrices from all steps of the sorting algorithm, $Q = \prod_{t=1}^{L} P_t$. As shown by (Petersen et al., 2021), $L = R$ of such steps are sufficient for efficient sorting. In the discrete case, for each column $i$, the permutation matrix has exactly one entry of 1, indicating the sequence element that should be placed in the $i$-th column. In the relaxed version, column values represent a distribution over possible sequence elements. In our case, a row $i$ of the distance matrix $D_s$ shows the distance of the $i$-th student feature to all the reference features.

With its sorting matrix $Q_i$, the $(r, k)$ element of this matrix can be viewed as the probability of a reference feature $r$ being the $k$-th nearest neighbor for the $i$-th feature. Hence, to maintain the order of nearest neighbors for every ROI-aligned patch feature, we compute $Q_i$ for every row of $D_s$ and $D_t$ and force these to be similar. This results in final matrices $Q^s = [Q_1^s, \ldots, Q_{N'}^s]$ and $Q^t = [Q_1^t, \ldots, Q_{N'}^t]$, which are used in the training loss.

**Training Loss.** After computing permutation matrices, we enforce similarity on the order of nearest neighbors for each of the aligned patch features using the cross-entropy loss. The loss for the permutation matrix of patch $i$ is defined as:

$$\mathcal{L}_{\text{CE}}(Q_i^t, Q_i^s) = -\sum_{j,k} Q_i^t(j, k) \log(Q_i^s(j, k)),$$

And the final training loss is defined as the sum of per-patch losses computed over all samples:

$$\mathcal{L}_{\text{NeCo}} = \sum_{i=1}^{N'} \mathcal{L}_{\text{CE}}(Q_i^t, Q_i^s)$$

This ensures, in a differentiable manner, that the order of nearest neighbors is consistent between the student and teacher features.

## 4 EXPERIMENTS

### 4.1 SETUP

**Benchmarked Methods.** We compare our method against state-of-the-art dense self-supervised learning methods, including CrIBo (Lebailly et al., 2023), Hummingbird (Balazevic et al., 2023), TimeT (Salehi et al., 2023), Leopart (Ziegler & Asano, 2022), and CrOC (Stegmüller et al., 2023) as well as baselines such ad DINO (Caron et al., 2021), iBOT (Zhou et al., 2022). To provide a more comprehensive evaluation, We also include the performance of DINOv2 enhanced with registers, referred to as DINOv2R (Oquab et al., 2023; Darcet et al., 2024), as it has demonstrated strong dense capabilities. Additionally, we benchmark our method against leading unsupervised semantic segmentation approaches such as COMUS (Zadaianchuk et al., 2022), DINOSAUR (Seitzer et al., 2022), DeepSpectral (Melas-Kyriazi et al., 2022), and MaskContrast (Van Gansbeke et al., 2021).

**Training.** We run our experiments on ViT-Small and ViT-Base with a patch size of 14. We start from various pretrained backbones, and use DINOv2 with registers unless otherwise noted. We post-pretrain these models for 25 COCO epochs on a single NVIDIA RTX A6000-46GB GPU, taking around 19 hours. For other training details, including detailed computational efficiency analysis, we refer readers to the Appendix A.

**Evaluation.** In all our evaluations, we discard the projection head, following previous works (Caron et al., 2021; Ziegler & Asano, 2022; Salehi et al., 2023), and directly use the spatial tokens from the Vision Transformer backbone. Scores in all experiments are reported as mean intersection over union (mIoU). We conduct four types of evaluations: linear segmentation fine-tuning with a $1 \times 1$

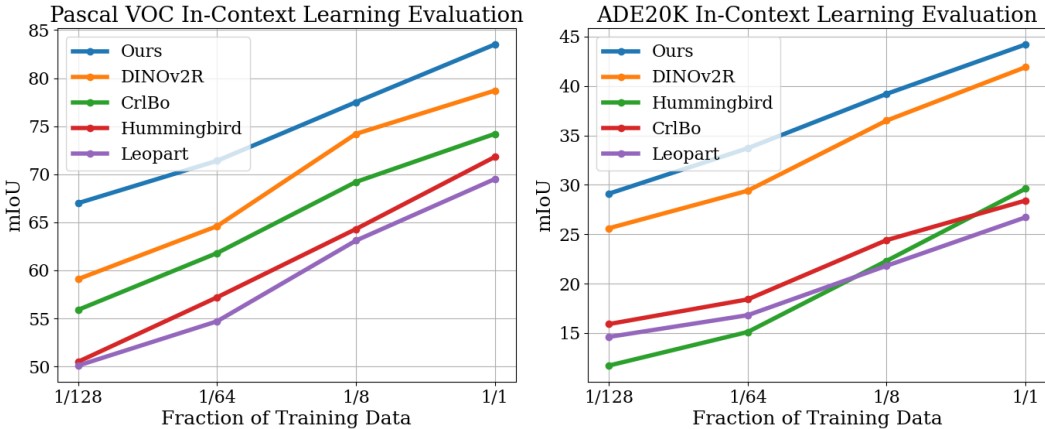

Figure 2: **In-context scene understanding benchmark.** Dense nearest neighbor retrieval performance is reported across various training data proportions on two scene-centric datasets, Pascal VOC and ADE20k. The retrieved cluster maps are compared with the ground truth using Hungarian matching (Kuang et al., 2021), and their mIoU score is reported. For all models, ViT-B16 is used except for DINOv2R and `NeCo`, where it is ViT-B14. For full tables, refer to Appendix B.3

convolution, end-to-end segmentation with the Segmenter head (Strudel et al., 2021), clustering and overclustering semantic segmentation (Ziegler & Asano, 2022; Salehi et al., 2023), and dense nearest neighbor retrieval (Balazevic et al., 2023). For clustering and overclustering, we apply K-Means to spatial tokens, setting $K$ to the number of ground truth objects and to high values like 300 and 500, as previously used (Ziegler & Asano, 2022; Salehi et al., 2023). We then extract object cluster maps and match them using Hungarian matching (Kuhn, 1955). For dense nearest neighbor retrieval, we follow the protocol from Balazevic et al. (2023), implemented in Pariza et al. (2024). For 3D understanding benchmark, We use the multiview feature consistency evaluation method proposed by El Banani et al. (2024), except all the images are resized to $224 \times 224$.

**Datasets.** We train our model on ImageNet-100 (Tian et al., 2020), Pascal VOC12 (Everingham et al., 2010), and COCO (Lin et al., 2014) for ablations and use COCO as our primary training dataset for all state-of-the-art comparisons. For evaluations, we use the validation sets of Pascal VOC12 (Everingham et al., 2010), COCO (Lin et al., 2014), ADE20k (Zhou et al., 2017), and Pascal Context (Mottaghi et al., 2014). For finetuning and feature transferability evaluations on COCO (Caesar et al., 2018), we train using a 10% split of the training set, while we use the full training splits of the other datasets. For the 3D understanding benchmark, we use Spair-71k (Min et al., 2019) dataset.

## 4.2 COMPARISON TO STATE-OF-THE-ART

In this section, we first compare the quality of frozen features learned through `NeCo` with state-of-the-art methods in in-context learning via nearest neighbor retrieval and unsupervised semantic segmentation tasks. Next, we demonstrate `NeCo`'s versatility by applying it to five different pretraining models and show it improves their dense features consistently. We then evaluate the transferability of our learned dense representations to other datasets by using linear head semantic segmentation and end-to-end fine-tuning with Segmentor (Strudel et al., 2021). Finally, we show that `NeCo` improves the 3D understanding of different vision models, as proposed in El Banani et al. (2024), using the multiview consistency experiment. We report the results of applying `NeCo` to more backbones in Appendix B.

**Visual In-Context Learning Evaluation.** We compare our approach to a recently proposed benchmark (Balazevic et al., 2023) that evaluates in-context reasoning in vision models. Unlike traditional linear segmentation methods, this evaluation does not require fine-tuning or end-to-end training. Instead, it creates validation segmentation maps by matching patch-level, feature-wise nearest neighbor similarities between validation images (queries) and training samples (keys). This method, inspired by NLP strategies, tests how well models learn tasks from a few examples. The results are presented

in Figure 2. As shown, NeCo outperforms prior state-of-the-art methods such as CrIBo and DINOv2R by 4% to 13% on Pascal and ADE20k across different fractions. The performance gap between NeCo and others increases, particularly in the data-efficiency regime. This improvement is due to our method's explicit enforcement of patch-level nearest neighbor consistency, resulting in higher-quality patch-level representations that remain effective even with fewer images. In contrast, other methods that promote image-level (Balazevic et al., 2023) or object-level (Lebailly et al., 2023) consistency force consistency between a pooled vector of patches, potentially leading to inadequate semantic patch-level representations, particularly in smaller datasets. Our method's ability to perform well with few images brings vision models one step closer to in-context learning style generalist reasoning. For complete tables and details of using other backbone variants, please refer to Appendix B.3. For visualizations please refer to Appendix C.

Table 1: **Frozen clustering-based evaluations.** *(a)* We evaluate the models by running $K$-means with various clustering granularities $K$ on the spatial features on two datasets. The resulting cluster maps are matched to the ground-truth by Hungarian matching, and the intersection is reported in mIoU. *(b)* Following previous works (Ziegler & Asano, 2022; Salehi et al., 2023), we post-process the resulting maps and report unsupervised semantic segmentation on Pascal VOC. Both tables use ViT-S with the patch size of 16, except for DINOv2R and NeCo, where it is 14.

(a) Clustering

| Method | Pascal VOC | | | COCO-Things | | |
|--------|------|-------|-------|------|-------|-------|
| | K=GT | K=300 | K=500 | K=GT | K=300 | K=500 |
| DINO | 4.3 | 13.9 | 17.3 | 5.4 | 18.8 | 19.2 |
| iBOT | 4.4 | 23.8 | 31.1 | 7.6 | 26.6 | 28.0 |
| CrOC | 3.4 | 16.4 | 20.0 | 4.9 | 14.7 | 18.1 |
| TimeT | 12.2 | 43.6 | 46.2 | 17.5 | 42.7 | 44.6 |
| DINOv2R | 12.2 | 46.7 | 49.5 | 12.3 | 38.9 | 41.2 |
| CrIBo | 18.3 | 51.3 | 54.5 | 14.5 | 46.0 | 48.3 |
| NeCo | **18.5** | **66.5** | **68.9** | **22.8** | **59.7** | **63.8** |

(b) Semantic segmentation

| Method | mIoU |
|--------|------|
| MaskConstrast | 35.1 |
| DINOv2R | 35.1 |
| DeepSpectral | 37.2 |
| DINOSAUR | 37.2 |
| Leopart | 41.7 |
| COMUS | 50.0 |
| NeCo | **55.6** |

**Frozen Clustering-based Evaluations.** Next, we evaluate the representation quality of our learned dense features across different objects in each dataset. Ideally, we expect that all patch features belonging to the same object, when clustered, will be assigned to the same cluster. If the learned representations are more fine-grained, such as learning object parts instead of whole objects (e.g., hands or faces instead of a person), they should consistently cover the same part across the entire dataset. To measure this, we extract dense features from all images and apply $K$-means clustering with various $K$ values to create cluster maps for each image. These cluster maps are then matched with the ground truth using Hungarian matching (Kuhn, 1955), and their mIoU is reported. For the first scenario, $K$ matches the number of ground truth objects. Additionally, to account for the second scenario, we also report performance in overclustering setups.

The results in Table 1a show that NeCo passes state-of-the-art by CrIBo by 14.5% on average across various datasets and metrics. Note that this gain is not due to the DINOv2R initialization, as it performs 4% lower than CrIBo on average. In Table 1b, we report clustering performance when $K$ matches the number of ground truth objects, with clustering applied solely to foreground patches extracted by methods used in Ziegler & Asano (2022); Salehi et al. (2023). We outperform other methods by at least 5.1% without relying on self-training, which requires training a separate segmentation head, as used in COMUS. For visualizations please refer to Appendix C.

**Linear Semantic Segmentation Evaluation.** In this experiment, we keep the pretrained backbone frozen and train a linear layer on top of the spatial features to solve a supervised semantic segmentation task. Bilinear interpolation is used to match the spatial feature resolution to the image size, enabling the application of pixel-wise cross-entropy loss. This setup provides a better evaluation of the pretrained models compared to end-to-end finetuning, where all learned parameters are overwritten. The results, reported in Table 2, show that NeCo surpasses CrIBo on all datasets by at least 10% and consistently outperforms DINOv2R, achieving gains of up to 4.8%. These significant improvements demonstrate that patches representing the same object or object part have higher similarities in feature space compared to other methods, as a simple linear layer can utilize these features for strong semantic segmentation.

Table 2: **Linear segmentation performance.** A linear segmentation head is trained on top of the frozen spatial features obtained from different feature extractors. We report the mIoU scores achieved on the validation sets of 4 different datasets.

| Method | Backbone | Params | COCO-Things | COCO-Stuff | Pascal VOC | ADE20K |
|--------|----------|--------|-------------|------------|------------|--------|
| DINO | ViT-S/16 | 21M | 43.9 | 45.9 | 50.2 | 17.5 |
| TimeT | ViT-S/16 | 21M | 58.2 | 48.7 | 66.3 | 20.7 |
| iBOT | ViT-S/16 | 21M | 58.9 | 51.5 | 66.1 | 21.8 |
| CrOC | ViT-S/16 | 21M | 64.3 | 51.2 | 67.4 | 23.1 |
| CrIBo | ViT-S/16 | 21M | 64.3 | 49.1 | 71.6 | 22.7 |
| DINOv2R | ViT-S/14 | 21M | 82.2 | 59.1 | 79.0 | 40.0 |
| NeCo | ViT-S/14 | 21M | **82.3** | **61.9** | **81.5** | **40.7** |
| DINO | ViT-B/16 | 85M | 55.8 | 51.2 | 62.7 | 23.6 |
| MAE | ViT-B/16 | 85M | 38.0 | 38.6 | 32.9 | 5.8 |
| iBOT | ViT-B/16 | 85M | 69.4 | 55.9 | 73.1 | 30.1 |
| CrIBo | ViT-B/16 | 85M | 69.6 | 53.0 | 73.9 | 25.7 |
| DINOv2R | ViT-B/14 | 85M | 84.8 | 59.3 | 80.2 | 43.0 |
| NeCo | ViT-B/14 | 85M | **86.4** | **64.1** | **84.4** | **46.5** |

**Compatibility with Differently Pretrained Backbones.** As shown in Table 3, our method is generalizable across various self-supervised learning initialization, improving them by roughly 4% to 30% across different metrics and datasets. Surprisingly, NeCo even enhances the performance of methods specifically designed for dense tasks, such as CrIBo, TimeT, and Leopart. For instance, CrIBo demonstrates a performance increase of approximately 5% in overclustering evaluations, which measures how fine-grained and semantic the representations learned during pretraining are. This indicates that NeCo applied to CrIBo can extract more discriminative features, leading to improved transfer performance, shown by 0.5% and 3.7% better linear classification performance on Pascal VOC and COCO-Things.

Table 3: **NeCo starting from different pretrainings.** We report frozen clustering and linear segmentation on Pascal VOC and COCO-Things. NeCo can considerably boost (↑) the performance of models with different initialization, showing our approach's generality. The backbone is ViT-S16.

| | Pascal VOC | | | | | | COCO-Things | | | | | |
|---|---|---|---|---|---|---|---|---|---|---|---|---|
| | *At Init* | | | +NeCo | | | *At Init* | | | +NeCo | | |
| Pretrain | K=GT | K=500 | Lin. | K=GT | K=500 | Lin. | K=21 | K=500 | Lin. | K=21 | K=500 | Lin. |
| iBOT (Zhou et al., 2022) | 4.4 | 31.1 | 66.1 | $15.4^{\uparrow 11.0}$ | $51.2^{\uparrow 20.1}$ | $68.6^{\uparrow 2.5}$ | 7.6 | 28.0 | 58.9 | $20.4^{\uparrow 12.8}$ | $52.8^{\uparrow 24.8}$ | $67.7^{\uparrow 8.8}$ |
| DINO (Caron et al., 2021) | 4.3 | 17.3 | 50.2 | $14.5^{\uparrow 10.2}$ | $47.9^{\uparrow 30.6}$ | $61.3^{\uparrow 11.1}$ | 5.4 | 19.2 | 43.9 | $16.9^{\uparrow 11.5}$ | $50.0^{\uparrow 30.8}$ | $62.4^{\uparrow 18.5}$ |
| TimeT (Salehi et al., 2023) | 12.2 | 46.2 | 66.3 | $17.9^{\uparrow 5.7}$ | $52.1^{\uparrow 5.9}$ | $68.5^{\uparrow 2.2}$ | 18.4 | 44.6 | 58.2 | $20.6^{\uparrow 2.2}$ | $54.3^{\uparrow 9.7}$ | $64.8^{\uparrow 6.6}$ |
| Leopart (Ziegler & Asano, 2022) | 15.4 | 51.2 | 66.5 | $21.0^{\uparrow 5.6}$ | $55.3^{\uparrow 4.1}$ | $68.3^{\uparrow 1.8}$ | 14.8 | 53.2 | 63.0 | $18.8^{\uparrow 4.0}$ | $53.9^{\uparrow 0.7}$ | $65.4^{\uparrow 2.4}$ |
| CrIBo (Lebailly et al., 2023) | 18.3 | 54.5 | 71.6 | $21.7^{\uparrow 3.4}$ | $59.6^{\uparrow 5.1}$ | $72.1^{\uparrow 0.5}$ | 14.5 | 48.3 | 64.3 | $21.1^{\uparrow 6.6}$ | $54.0^{\uparrow 5.7}$ | $68.0^{\uparrow 3.7}$ |

**End-to-End Full-Finetuning Evaluation.** One advantage of self-supervised pretraining is the ability to transfer learned general semantic features to specialized downstream tasks, improving performance in an end-to-end finetuning setup. We evaluate this capability of NeCo by adding a transformer-based decoder from Segmenter (Strudel et al., 2021) on top of the feature extractor and finetuning the entire network for semantic segmentation. The backbone's spatial features are fed into a transformer decoder along with $K$ learnable class tokens. These class and spatial tokens are projected onto each other to obtain patch-level predictions, which are then upsampled to match the input image size, enforcing pixel-wise cross-entropy loss. We report the mIoU scores achieved on Pascal VOC, Pascal Context, COCO-Stuff, and ADE20k in Table 4. Despite all parameters being adapted, the results show that NeCo learns superior features, leading to better performance in downstream tasks, outperforming CrIBo by at least 3.4% across different datasets and backbones. Notably, while DINOv2R has demonstrated strong transfer results in various tasks, including semantic segmentation, due to being trained on a massive dataset of 142M images and using a combination of dense and classification losses, NeCo surpasses even this. By training only 19 GPU-hours on COCO, which is a fraction of the original compute, we obtain consistent gains of up to 1.7%, setting a new state-of-the-art.

**Multiview Feature Consistency Evaluation.** This evaluation assesses the 3D understanding of models through geometric correspondence estimation, aiming to measure the consistency of feature

Table 4: **Evaluation of Full Fine-Tuning with Segmenter.** Various backbones pre-trained with different self-supervised learning methods are fine-tuned using Segmenter (Strudel et al., 2021). The table shows the mIoU scores obtained on validation sets across 4 different datasets.

| Method | Backbone | Params | **Pascal Context** | **Pascal VOC** | **COCO-Stuff** | **ADE20K** |
|--------|----------|--------|--------------------|----------------|----------------|-----------|
| DINO | ViT-S/16 | 21M | 46.0 | 80.3 | 43.2 | 43.3 |
| CrOC | ViT-S/16 | 21M | 46.0 | 80.9 | 42.9 | 42.8 |
| TimeT | ViT-S/16 | 21M | 47.4 | 80.4 | 43.1 | 43.5 |
| CrIBo | ViT-S/16 | 21M | 49.3 | 82.3 | 43.9 | 45.2 |
| DINOv2R | ViT-S/14 | 21M | 59.3 | 86.2 | 46.9 | 48.6 |
| NeCo | ViT-S/14 | 21M | **59.7** | **87.0** | **47.3** | **48.9** |
| DINO | ViT-B/16 | 85M | 45.8 | 82.2 | 44.4 | 45.0 |
| MAE | ViT-B/16 | 85M | 47.9 | 82.7 | 45.5 | 46.4 |
| CrIBo | ViT-B/16 | 85M | 49.2 | 83.4 | 44.6 | 46.0 |
| DINOv2R | ViT-B/14 | 85M | 62.4 | 86.0 | 48.8 | 52.3 |
| NeCo | ViT-B/14 | 85M | **63.0** | **87.7** | **49.3** | **52.5** |

Table 5: **Multiview feature consistency results on SPair-71k, Recall@0.01.** We use the evaluation method proposed by El Banani et al. (2024). NeCo improves the results of DINO models by roughly 1.8% for 3D understanding, measured by multiview feature consistency.

| | ✈ | 🚲 | 🐦 | 🛥 | 🚌 | 🚗 | 🐱 | 🏛 | 🐕 | 🍴 | 🐎 | 🐏 | 🦜 | 🪴 | 🐑 | 🚆 | 📺 | 🖥 | Avg |
|---|---|---|---|---|---|---|---|---|---|---|---|---|---|---|---|---|---|---|---|
| DINO-B16 | **26.7** | 14.9 | 35.4 | 15.6 | 20.6 | 19.3 | **18.1** | 33.7 | **11.2** | 19.4 | **23.9** | 16.3 | 16.3 | 18.8 | 11.1 | 12.9 | 29.5 | 10.1 | 19.6 |
| + NeCo | 26.0 | **17.5** | **36.7** | **16.9** | **22.0** | **23.5** | 18.0 | 33.5 | 10.6 | **21.0** | 21.6 | **20.2** | **20.6** | **19.3** | **13.7** | **13.9** | **36.3** | **12.5** | **21.3** |
| DINOV2R-B14 | 31.2 | **34.4** | 56.7 | **23.2** | 26.1 | 29.4 | **37.5** | 51.3 | 20.8 | 36.6 | 36.7 | 31.7 | **26.2** | 29.4 | 15.4 | **26.5** | 39.2 | 11.9 | 32.5 |
| +NeCo | **40.6** | 25.7 | **58.0** | 20.4 | **37.1** | 33.3 | 37.2 | **56.2** | **24.6** | **37.7** | **40.8** | **33.1** | 23.7 | **35.1** | **22.8** | 23.9 | **45.4** | **22.4** | **34.3** |

representations across multiple views of the same scene. By identifying matching points in different images without additional model training, it provides a direct evaluation of 3D feature quality (El Banani et al., 2024). The evaluation is done on a large-scale dataset called Spair-71k (Min et al., 2019). As Table 5 shows, NeCo can boost the performance of DINO models by roughly 10% on average across different categories. This demonstrates that the proposed method extends beyond enhancing semantic segmentation and has a broader impact on vision foundation models by improving their overall spatial understanding

## 4.3 ABLATION STUDIES

Here, we examine the essential parameters of our method by training NeCo on Pascal VOC12 and ADE20k. We assess its ability to perform linear segmentation and in-context scene understanding using the frozen representations learned with each set of parameters. For in-context scene understanding evaluations, we use $\frac{1}{128}$ fraction of the training data and reduce the spatial dimension to $448^2$. The number of training epochs for linear segmentation evaluations is set to 20 epochs. For more ablations, including the effect of training epochs and sorting hyperparameters, refer to Appendix B.

**Patch Selection Approach.** We demonstrate the effect of selecting patches from the foreground, background, or both in Table 6a. Foreground patches are selected using the attention map averaged across heads. Our results indicate that selecting patches from the foreground gives 1% better results compared to the background selection in 3 out of 4 metrics. However, the performance peaks when we select patches from both locations. This improvement can be attributed to the use of scene-centric images for training, where the background often contains meaningful objects that contribute to enhanced performance.

**Utilizing a Teacher.** We ablate the role of teacher-student architecture in Table 6b. As shown, employing a teacher network updated by exponential moving average can significantly improve the performance across all the metrics by 8% to 20%. This is consistent with the previous works (Caron et al., 2021; Grill et al., 2020), which reported a more stable training process when the teacher-student architecture is employed.

**Nearest Neighbor Selection Approach.** We evaluate the influence of picking nearest neighbors from the same image (intra) or different batch images (inter) and find that selecting patches across images consistently boosts performance by roughly 0.4% to 1% across different metrics. The higher diversity

Table 6: **Ablations of the key parameters of our method.** We evaluate the models by training a linear layer on top of the frozen representations (Lin.) or using the in-context (IC) evaluation of Balazevic et al. (2023) using the validation images for PascalVOC12 and ADE20k.

(a) Patch selection

| Location | Pascal | | ADE20K | |
|---|---|---|---|---|
| | Lin. | IC | Lin. | IC |
| backg. | 78.4 | 60.5 | 35.8 | 20.6 |
| foreg. | 78.4 | 61.6 | 36.8 | 21.5 |
| both | **78.9** | **62.0** | **37.3** | **21.7** |

(b) Use of EMA Teacher

| Teacher | Pascal | | ADE20K | |
|---|---|---|---|---|
| | Lin. | IC | Lin. | IC |
| ✗ | 70.4 | 42.6 | 28.3 | 15.9 |
| ✓ | **78.9** | **62.0** | **37.3** | **21.7** |

(c) Num Neighbors

| Num | Pascal | | ADE20K | |
|---|---|---|---|---|
| | Lin. | IC | Lin. | IC |
| 4 | 74.4 | 54.8 | 35.2 | 19.7 |
| 8 | 76.8 | 61.1 | 36.3 | 20.2 |
| 16 | 77.7 | 60.9 | 36.7 | 20.9 |
| 32 | 78.1 | 61.3 | 37.1 | 21.4 |
| All | **78.9** | **62.0** | **37.3** | **21.7** |

(d) Training dataset

| Dataset | Pascal | | ADE20K | |
|---|---|---|---|---|
| | Lin. | IC | Lin. | IC |
| IN-100 | 76.7 | 55.8 | 34.9 | 18.7 |
| Pascal | 77.9 | 60.6 | 36.4 | 20.8 |
| COCO | **78.9** | **62.0** | **37.3** | **21.7** |

(e) Batch Size

| Batch Size | Pascal | | ADE20K | |
|---|---|---|---|---|
| | Lin. | IC | Lin. | IC |
| 1 | 76.0 | 60.0 | 35.4 | 20.1 |
| 4 | 76.2 | 60.2 | 35.6 | 20.2 |
| 8 | 76.8 | 61.1 | 36.3 | 20.8 |
| 16 | 77.7 | 61.2 | 36.7 | 21.1 |
| 32 | 78.2 | 61.4 | 37.1 | 21.4 |
| 64 | **78.9** | **62.0** | **37.3** | **21.7** |

of patches involved in the latter approach likely accounts for this improvement (see Appendix B.7 for full tables).

**Training Dataset.** Table 6d presents the impact of the training dataset based on the ImageNet-100, Pascal, and COCO datasets. ImageNet-100 comprises relatively simple images with few objects, whereas Pascal and COCO feature more complex scenes with multiple objects. Our method shows consistent improvements when trained on multi-object datasets, achieving a performance increase of 2% to 7% on COCO compared to ImageNet-100. This improvement is due to the greater quantity and diversity of objects per batch in multi-object scenes, which provide stronger learning signals by requiring discrimination against a higher number of objects. Notably, the additional performance boost we observe from finetuning DINOv2R on Pascal—despite it already being trained on this dataset (Oquab et al., 2023)—further underscores the efficacy of our proposed loss function.

**Sorting Algorithm.** We ablate the effect of changing the sorting algorithm and find that our method maintains strong performance across various approaches, achieving the best results with Bitonic sorting, which slightly outperforms the alternatives on average. Additionally, we investigate the absence of a sorting component, which leads to deteriorated performance. The complete tables for this study, along with an ablation on the sorting steepness parameter, are provided in Appendix B.7, as our method is robust to variations in sorting parameters.

**Batch Size.** Table 6e examines how batch size affects performance. Smaller batch sizes provide marginal gains, but larger batch sizes show more significant improvements, indicating that performance could be further improved with batch sizes exceeding 64.

**Number of Neighbors.** As detailed in the method section, we use a differentiable sorting algorithm to compute and sort the distances between each patch and others. The ablation study in Table 6c evaluates selecting the top $K$ distances instead of all. Results show that incorporating more neighbors improves performance, but beyond a threshold (e.g., 32), the effect diminishes, indicating robustness against this hyperparameter.

## 5 CONCLUSION

In this work, we propose Patch Nearest Neighbor Consistency as a new method for dense post-pretraining of self-supervised backbones. By applying our method to the many backbones including the DINOv2-registers model, we improve upon these models by a large margin for frozen clustering, semantic segmentation, full finetuning, and 3D understanding, setting several new state-of-the-art performances.

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

## A EXPERIMENTAL SETUP

### A.1 DENSE POST-PRETRAINING

**Implementation Framework**  Our model is implemented in Python, using Torch (Paszke et al., 2019) and PyTorch Lightning (Falcon & team, 2019).

**Datasets**  Our pretraining datasets consist of COCO (Caesar et al., 2018) and ImageNet-100 subset of the original Imagenet (Russakovsky et al., 2015). COCO contains approximately 118,000 scene-centric images, whereas ImageNet-100k includes around 100k object-centric images.

**Data Augmentations**  Our Data augmentations are the same as in Ziegler & Asano (2022). More specifically, we use: random color-jitter, Gaussian blur, grayscale and multi-crop augmentations. Similarly, the global crop's resolution is 224x224 and the local crop's resolution is 96x96, for the all the experiments except when working with Dinov2 where we use 518x518 for global crops and 98x98 for local crops. The only place where the resolution is different for NeCo using DINOv2 is in the ablation studies, where we use 224x224 for global crops and 98x98 for local crops. Furthermore, our generated global and local crops have the constraint that they intersect at least by $1\%$ of the original image size.

**Network Architecture**  For our backbone, we employ vision transformers. More specifically, we train on ViT-Small and ViT-Base (Dosovitskiy et al., 2020). Moreover, following Caron et al. (2021); Grill et al. (2020), we use a student-teacher setup where the teacher weights are updated by the exponential moving average of the student weights.

**Registers in Dinov2**  We report results on *DINOv2R* (Darcet et al., 2024), *DINOv2XR*, and *DINOv2* (Oquab et al., 2023). For DINOv2XR, we remove the registers to match the input patch structure of the original DINOv2 model. DINOv2XR serves as an interesting experiment to evaluate whether removing registers, thus simplifying the architecture to align with DINOv2, affects performance significantly. However, all our primary results and comparisons focus on DINOv2R.

**Projection Head**  Following Caron et al. (2021), the projection head consists of three linear layers with hidden dimensionality of 2048, a Gaussian error linear units as activation function (Hendrycks & Gimpel, 2016), and an output dimensionality of 256.

**Dense Image Representation Alignment of Crops**  Following Ziegler & Asano (2022), due to the distinction between global and local crops, after projecting the dense spatial output to a lower space, the alignment step is applied on the dense image representations to bring them to a fixed spatial resolution of size of 7x7 during training. This ensures that the local and global crop feature maps have the same size and that they correspond to each other. The alignment is done using region of interest alignment (roi align) (He et al., 2017).

**Optimization**  We train both network sizes with a cosine learning rate schedule going down to $0$ over 25 training epochs, except for the ablation studies where we use 10 epochs. The initial projection head learning rate is $1e{-}4$ for all the experiments, whereas the backbone's learning rate is $1e{-}5$, with the exception of being $1e{-}6$ when applying our method on Dinov2. The exponential moving average for updating the teacher's weights is adapted with a cosine schedule starting at $0.9995$ and going up to $1$. We use Adam Optimizer (Kingma & Ba, 2017) with a cosine weight decay schedule.

**Differentiable Sorting Networks**  By default we use the Bitonic Differentiable Sorting Networks (Petersen et al., 2021) and the steepnesses (i.e., inverse temperatures) used for the network are 100 for the Student and 100 for the teacher. All the other parameters remain as the default ones; i.e., we use the $logistic_\phi$ function with a $\lambda = 0.25$ for the interpolation of numbers in the differentiable sorting algorithms.

Table 7: **Post-Training Configurations for Different Experimental Setups used in this work.** This set of tables shows all the different Post-Training parameter configurations used for post-training models for various experiments. The last row of each configuration table shows the GPU hours that are approximately used for post-training the model with the respective configuration. The rest of the parameters are the same and are as explained in Appendix A.1. Note, that the $^*$ for the configuration in Table 7a, denotes that the model is not eventually trained for 25 epochs but instead stopped after around the GPU hours mentioned since the model does not provide significant improvements afterwards (at around the 7th epoch). Last DINOv2R-XR means DINOv2R but excluding registers from the checkpoint().

(a) DINOv2R Best

| Param | Value |
|---|---|
| Global Crop | 518 |
| Local Crop | 98 |
| Backbone LR | 1e−6 |
| Head LR | 1e−4 |
| Max Epochs | 25* |
| Drop Registers | No |
| GPU Hours | 18.8 |

(b) DINOv2R-XR Best

| Param | Value |
|---|---|
| Global Crop | 518 |
| Local Crop | 98 |
| Backbone LR | 1e−6 |
| Head LR | 1e−4 |
| Max Epochs | 25* |
| Drop Registers | Yes |
| GPU Hours | 18.8 |

(c) DINOv2 Best

| Param | Value |
|---|---|
| Global Crop | 518 |
| Local Crop | 98 |
| Backbone LR | 1e−6 |
| Head LR | 1e−4 |
| Max Epochs | 25* |
| Drop Registers | N/A |
| GPU Hours | 18.8 |

(d) DINOv2R-XR Ablations

| Param | Value |
|---|---|
| Global Crop | 224 |
| Local Crop | 98 |
| Backbone LR | 1e−6 |
| Head LR | 1e−4 |
| Max Epochs | 10 |
| Drop Registers | Yes |
| GPU Hours | 7.2 |

(e) All others than DINOv2R

| Param | Value |
|---|---|
| Global Crop | 224 |
| Local Crop | 96 |
| Backbone LR | 1e−5 |
| Head LR | 1e−4 |
| Max Epochs | 25 |
| Drop Registers | N/A |
| GPU Hours | 17.5 |

Table 8: **The Post-Training Configuration used for each experiment table.**

| Table | Configuration |
|---|---|
| 1 | 7a |
| 2 | 7a |
| 3 | 7e |
| 4 | 7a |
| 5 | 7a for DINOv2R / 7e for DINO |

| Table | Configuration |
|---|---|
| 6 | 7d |
| 9 | 7a |
| 10 | 7a, 7b, 7c |
| 11 | 7a, 7b, 7c |
| 12 | 7e for DINO / 7a, 7b, 7c for rest |

| Table | Configuration |
|---|---|
| 13 | 7e |
| 14 | 7e |
| 15 | 7a, 7b, 7c |
| 16 | 7d |
| 17 | 7e |

## A.2 EVALUATION SETUP

**Visual In-Context Learning** The Dense Nearest Neighbor Retrieval Evaluation is a retrieval-based scene understanding evaluation introduced by Balazevic et al. (2023). Its goal is to assess the scene understanding capabilities of a dense image encoder. It works as follows:

1. **Memory Bank Construction**: Using a dataset of images and their dense annotations, two memory banks are created. One memory bank stores image patch features extracted from the spatial output of a dense encoder applied to the training images. The other memory bank stores the corresponding patch labels from the dataset annotations.

2. **Query Processing**: For an image from the validation split, the spatial output of the dense image encoder is processed. For each patch representation in this output, the $k$ nearest neighbors are identified from the memory bank of features. The labels of these nearest neighbors are then combined to construct the query's label.

3. **Comparison**: After constructing the annotation for the entire image, it is compared with the ground truth annotation.

Due to the unavailability of the original implementation by Balazevic et al. (2023), we use the open implementation from Pariza et al. (2024). This implementation aligns with the original authors' description and details, including the use of the ScaNN Library (Guo et al., 2020) for efficient nearest neighbor retrieval. We adhere to the setup from the Hummingbird Model authors (Balazevic et al., 2023) for our experiments. We use a memory size of 10,240,000 and configure ScaNN with 30 nearest neighbors, consistent with the evaluation of the Hummingbird model on this memory size.

The final results are reported as mean Intersection over Union (mIoU) on four different fractions of two datasets: Pascal VOC 2012 (Everingham et al.) and ADE20K (Zhou et al., 2017). The sub-sampling factors are 1, 8, 64, or 128. For factors greater than 1, results are averaged over five different seeds. These dataset subsets are created by uniformly and randomly selecting a unique set of images from the training split, ensuring an approximately equal number of distinct images for each annotation label. For example, for the 1/128 fraction of the Pascal VOC 2012 dataset, we would collect around 83 images, ensuring each of the 20 labels (excluding the background) appears in at least 4 different images in the subset.

**Overclustering.** For the Overclustering experiment, following Ziegler & Asano (2022), we run $K$-Means (using faiss (Johnson et al., 2019)) on all spatial tokens from our backbone (i.e., with the projection head discarded) for a given dataset. We then group the clusters to the ground-truth classes of the dataset by applying greedy matching to the pixel-level precision and then run Hungarian matching (Kuhn, 1955) on the combined cluster maps, which makes the evaluation metric permutation-invariant (Ji et al., 2019). We use a crop size of 448x448 for the input images, and overclustering is applied on downsampled 100x100 masks in order to speed up the Hungarian matching. The final results are reported as an average of mean Intersection over Union (mIoU) over five different seeds on four different datasets: COCO-Thing and COCO-Stuff (Caesar et al., 2018), Pascal VOC 2012 (Everingham et al.), and ADE20K (Zhou et al., 2017).

**Linear segmentation** For linear segmentation, we closely follow the setup from Leopart (Ziegler & Asano, 2022). Concisely, we take 448x448 images, encode them with our backbone to get the spatial outputs, apply bilinear interpolation to match the mask resolution, and finally apply a linear head to obtain the segmentation predictions. These predictions are then compared with the ground truth segmentation masks and trained via cross-entropy loss.

For training the linear head, we downsample the segmentation masks to 100x100 to increase training speed. We use Stochastic Gradient Descent with a weight decay of 0.0001, a momentum of 0.9, and a step learning rate scheduler. We found that a learning rate of 0.01 works quite well for the backbone models we evaluated and our setup. We fine-tune the linear heads for 20 epochs.

Moreover, we train and evaluate linear heads on four versions of datasets: Pascal VOC 2012 (Everingham et al.), subsets of COCO-Thing and COCO-Stuff (explained in Appendix A), and ADE20K (Zhou et al., 2017).

**Segmenter Finetuning** Following the evaluation setup from Lebailly et al. (2023), we finetune our backbones and the transformer-based decoder from Segmenter (Strudel et al., 2021) in an end-to-end manner. We use the Segmenter implementation available within the MMSegmentation Library (MMSegmentation Contributors, 2020).

The performance metric used here is the *mIoU score*, reported on four different datasets: Pascal Context (Everingham et al., 2010), Pascal VOC 2012 (Everingham et al.), COCO-Stuff 164K (Caesar et al., 2018), and ADE20K (Zhou et al., 2017). The crop size used is 512×512. For the DINOv2 model and our method on it, we apply zero padding around the image of 512×512 to bring it to the size of 518×518.

The remaining configurations follow Lebailly et al. (2023). For the ADE20K and COCO-Stuff 164K datasets, we use 160k iterations, and for Pascal VOC 2012 and Pascal Context, we use 80k iterations, all with an *eta_min* of $0.1 \cdot lr$. We use the Adam optimizer (Kingma & Ba, 2017) and for each pretraining method and dataset, we experiment with four different learning rates ($8 \times 10^{-5}, 3 \times 10^{-5}, 1 \times 10^{-5}, 8 \times 10^{-6}$) before reporting the highest mIoU score.

**Fully unsupervised semantic segmentation**    To better evaluate the scene understanding abilities of our method, we also evaluate it using the Fully Unsupervised Semantic Segmentation Evaluation method (Ziegler & Asano, 2022). This evaluation consists of two parts: Cluster-based Foreground Extraction (CBFE) and Overclustering with Community Detection (CD).

The CBFE clusters the spatial outputs of a model over a dataset and assigns each cluster as background (*bg*) or foreground (*fg*). The separation of foreground and background clusters is facilitated by attention maps from a Vision Transformer, which provide cues for the fg/bg distinction. We construct the final hard fg-bg assignment by averaging the attention heads, applying Gaussian filtering with a 7x7 kernel size, and retaining 70% of the attention mass to obtain the final binary mask. The rest of the configurations remain the same as the original setup (Ziegler & Asano, 2022).

The CD metric (Ziegler & Asano, 2022) exploits local co-occurrence statistics among clusters to identify and categorize objects. This approach uses no labels for categorizing semantic parts; it simply finds local co-occurrence of clusters in an image by utilizing an information-theoretic definition of network communities. Our configurations for the CD evaluation remain the same as in Leopart (Ziegler & Asano, 2022).

We use the implementation from Leopart (Ziegler & Asano, 2022) and apply CBFE and CD on the non-augmented (*train*) split of Pascal VOC 2012 (Everingham et al.), and evaluate on its full validation set. For CD, we report the best results over 10 seeds obtained from a hyper-parameter search, leading to our best parameters for CD+CBFE: *weight_threshold* = 0.07, *markov_time* = 1.2, and *k_community* = 189.

# B    ADDITIONAL EXPERIMENTS

## B.1    UNSUPERVISED SEMANTIC SEGMENTATION

In Table 1b, we show the contribution of each component to the final clustering evaluation gains on Pascal VOC 2012 with 21 clusters for DINOv2R ViT-small. Starting with our method `NeCo`, we then apply Clustering Based Foreground Extraction (CBFE), and finally community detection (CD). We observe that CBFE provides the largest boost (23.3%) due to the high quality of our overclustering maps, as also shown in Table 1a. While CD contributes a more modest increase (13.8%), which is about half as much as CBFE, combining both CBFE and CD results in a significant improvement, bringing the overall gain to 55.1%, compared to the initial 17.8%.

Table 9: Component contributions. We show the gains that each individual component brings for PVOC segmentation and K=21.

|  | mIoU |
| --- | --- |
| DINOv2R | 12.2 |
| + `NeCo` | 17.8 (+5.6%) |
| + CBFE | 41.5 (+23.7%) |
| + CD | 55.6 (+13.9%) |

## B.2    FULL CLUSTERING TABLES

In Table 10, we show full clustering results shown by Table 1b.

## B.3    FULL VISUAL IN-CONTEXT LEARNING TABLES

We show the results shown by Figure 2 in Table 11. To provide a more comprehensive comparison, we also evaluate our method against SelfPatch (Yun et al., 2022), a finetuning approach with a similar aim of enhancing dense representations.

Table 10: **Clustering evaluation performance.** K-means with various clustering granularity $K$ is applied to the spatial features obtained from different feature extractors on two datasets. The resulting cluster maps are matched to the ground truth by Hungarian matching (Kuhn, 1955), and the intersection is reported in mIoU.

| Method | Backbone | Params | Pascal VOC | | | COCO-Things | | |
|---|---|---|---|---|---|---|---|---|
| | | | K=GT | K=300 | K=500 | K=GT | K=300 | K=500 |
| DINO (Caron et al., 2021) | ViT-S/16 | 21M | 4.3 | 13.9 | 17.3 | 5.4 | 18.8 | 19.2 |
| iBOT (Zhou et al., 2022) | ViT-S/16 | 21M | 4.4 | 23.8 | 31.1 | 7.6 | 26.6 | 28.0 |
| CrOC (Stegmüller et al., 2023) | ViT-S/16 | 21M | 3.4 | 16.4 | 20.0 | 4.9 | 14.7 | 18.1 |
| TimeT (Salehi et al., 2023) | ViT-S/16 | 21M | 12.2 | 43.6 | 46.2 | 17.5 | 42.7 | 44.6 |
| DINOv2R (XR) (Oquab et al., 2023) | ViT-S/14 | 21M | 12.2 | 46.7 | 49.5 | 12.3 | 38.9 | 41.2 |
| CrIBo (Lebailly et al., 2023) | ViT-S/16 | 21M | 18.3 | 51.3 | 54.5 | 14.5 | 46.0 | 48.3 |
| DINOv2 | ViT-S/14 | 21M | 17.1 | 53.5 | 58.2 | 19.7 | 51.6 | 53.8 |
| DINOv2R | ViT-S/14 | 21M | 18.0 | 59.1 | 64.5 | 20.1 | 55.1 | 59.2 |
| NeCo (DINOv2) | ViT-S/14 | 21M | 15.5 | 50.4 | 57.7 | 20.0 | 56.0 | 60.0 |
| NeCo (DINOv2R) | ViT-S/14 | 21M | **18.5** | 66.5 | 68.9 | **22.8** | 59.7 | 63.6 |
| NeCo(DINOv2XR) | ViT-S/14 | 21M | 17.8 | 69.4 | 72.6 | 18.2 | 61.2 | 64.5 |
| MAE | ViT-B/16 | 85M | 3.5 | 6.0 | 7.4 | 6.9 | 9.2 | 10.1 |
| DINO | ViT-B/16 | 85M | 5.3 | 19.6 | 23.9 | 6.4 | 19.1 | 21.2 |
| iBOT | ViT-B/16 | 85M | 6.5 | 29.0 | 34.0 | 7.2 | 26.4 | 30.5 |
| DINOv2R (XR) | ViT-B/14 | 85M | 14.4 | 47.7 | 50.5 | 12.4 | 30.9 | 33.5 |
| CrIBo | ViT-B/16 | 85M | **18.9** | 56.9 | 56.8 | 16.2 | 43.1 | 44.5 |
| DINOv2 | ViT-B/14 | 85M | 15.5 | 52.8 | 56.9 | 22.6 | 54.0 | 54.9 |
| DINOv2R | ViT-B/14 | 85M | 21.4 | 62.2 | 64.6 | 23.1 | 55.2 | 57.8 |
| NeCo (DINOv2) | ViT-B/14 | 85M | 17.2 | 66.8 | 71.1 | **23.7** | 62.2 | 63.1 |
| NeCo (DINOv2R) | ViT-B/14 | 85M | 17.2 | 72.2 | 71.9 | 17.3 | 62.1 | 64.6 |
| NeCo (DINOv2XR) | ViT-B/14 | 85M | 18.6 | 64.2 | 71.8 | 13.3 | 61.3 | **65.5** |

Table 11: **In-context scene understanding benchmark.** Dense nearest neighbor retrieval performance is reported across various training data proportions on two scene-centric datasets, ADE20k and Pascal VOC. The retrieved cluster maps are compared with the ground truth using Hungarian matching (Kuhn, 1955), and their mIoU score is reported.

| Method | Backbone | Params | ADE20K | | | | Pascal VOC | | | |
|---|---|---|---|---|---|---|---|---|---|---|
| | | | 1/128 | 1/64 | 1/8 | 1/1 | 1/128 | 1/64 | 1/8 | 1/1 |
| DINO | ViT-S/16 | 21M | 9.5 | 11.0 | 15.0 | 17.9 | 26.4 | 30.5 | 41.3 | 48.7 |
| SelfPatch | ViT-S/16 | 21M | 10.0 | 10.9 | 14.7 | 17.7 | 28.4 | 32.6 | 43.2 | 50.8 |
| CrOC | ViT-S/16 | 21M | 8.7 | 10.8 | 15.2 | 17.3 | 34.0 | 41.8 | 53.8 | 60.5 |
| TimeT | ViT-S/16 | 21M | 12.1 | 14.1 | 18.9 | 23.2 | 38.1 | 43.8 | 55.2 | 62.3 |
| Leopart | ViT-S/16 | 21M | 12.9 | 14.8 | 19.6 | 23.9 | 44.6 | 49.7 | 58.4 | 64.5 |
| CrIBo | ViT-S/16 | 21M | 14.6 | 17.3 | 22.7 | 26.6 | 53.9 | 59.9 | 66.9 | 72.4 |
| DINOv2XR | ViT-S/14 | 21M | 19.6 | 22.8 | 30.1 | 35.9 | 53.0 | 57.9 | 68.2 | 75.0 |
| DINOv2 | ViT-S/14 | 21M | 22.4 | 25.8 | 33.6 | 38.9 | 55.7 | 61.8 | 72.4 | 77.0 |
| DINOv2R | ViT-S/14 | 21M | **23.7** | 27.1 | 33.9 | 39.5 | 60.1 | 65.7 | 74.5 | 78.8 |
| NeCo (DINOv2) | ViT-S/14 | 21M | 22.1 | 25.2 | 32.9 | 38.0 | 59.8 | 64.6 | 73.4 | 78.6 |
| NeCo (DINOv2R) | ViT-S/14 | 21M | **23.7** | **27.2** | **34.7** | **40.9** | 65.6 | 70.1 | **76.8** | **80.7** |
| NeCo (DINOv2XR) | ViT-S/14 | 21M | **23.7** | **27.2** | 34.0 | 39.8 | **66.5** | **70.3** | 76.3 | 80.2 |
| MAE | ViT-B/16 | 85M | 10.0 | 11.3 | 15.4 | 18.6 | 3.5 | 4.1 | 5.6 | 7.0 |
| DINO | ViT-B/16 | 85M | 11.5 | 13.5 | 18.2 | 21.5 | 33.1 | 37.7 | 49.8 | 57.3 |
| Leopart | ViT-B/16 | 85M | 14.6 | 16.8 | 21.8 | 26.7 | 50.1 | 54.7 | 63.1 | 69.5 |
| Hummingbird | ViT-B/16 | 85M | 11.7 | 15.1 | 22.3 | 29.6 | 50.5 | 57.2 | 64.3 | 71.8 |
| CrIBo | ViT-B/16 | 85M | 15.9 | 18.4 | 24.4 | 28.4 | 55.9 | 61.8 | 69.2 | 74.2 |
| DINOv2XR | ViT-B/14 | 85M | 22.1 | 25.8 | 33.2 | 38.7 | 51.8 | 58.9 | 70.6 | 77.3 |
| DINOv2 | ViT-B/14 | 85M | 22.1 | 25.2 | 32.3 | 37.9 | 54.1 | 60.5 | 71.5 | 76.7 |
| DINOv2R | ViT-B/14 | 85M | 25.6 | 29.4 | 36.5 | 41.9 | 59.1 | 64.6 | 74.2 | 78.7 |
| NeCo (DINOv2) | ViT-B/14 | 85M | 26.6 | 31.1 | 38.9 | 43.4 | 68.4 | 72.8 | 79.6 | 82.8 |
| NeCo (DINOv2R) | ViT-B/14 | 85M | 27.8 | 32.1 | **39.7** | **44.5** | **69.0** | **73.1** | **79.8** | 82.9 |
| NeCo (DINOv2XR) | ViT-B/14 | 85M | **29.1** | **33.7** | 39.2 | 44.2 | 67.0 | 71.4 | 77.5 | **83.5** |

## B.4 OBJECT DETECTION WITH VITDET

We report the performance of ViTDet (Li et al., 2022) on COCO dataset for ViT-S with DINO, DINOv2, and DINOv2R backbones and compare them with the `NeCo` finetuned versions. Our method consistently improves all the DINO family backbones, as shown by Table 12. In particular, `NeCo` improves validation Box AP and Mask AP by 2.4% and 3.3%, respectively, over DINOv2, highlighting the versatility of our approach

Table 12: **Object detection performance with ViTDet.** Although `NeCo` is trained only for 19 GPU hours, it can still improve the performance of DINO backbones on average across the specified measures.

| Backbone | Epochs | Val Box AP | Val Mask AP |
|---|---|---|---|
| Dino | 12 | 42.9 | 38.6 |
| + NeCo | 12 | **43.0** | **38.7** |
| Dinov2XR | 12 | 42.5 | 36.7 |
| + NeCo | 12 | **42.6** | **36.7** |
| Dinov2 | 12 | 45.7 | 40.0 |
| + NeCo | 12 | **48.1** | **41.4** |
| Dinov2R | 12 | 47.9 | 41.1 |
| + NeCo | 12 | **48.3** | **41.4** |

## B.5 APPLYING NECO TO VISION-LANGUAGE FOUNDATION MODELS

We present the results for CLIP (Radford et al., 2021) and SigLIP (Zhai et al., 2023) on linear segmentation and visual in-context learning in Table 14 and Table 13. As shown in the tables above, `NeCo` improves the performance of both CLIP and SigLIP by approximately 12% to 37% across various benchmarks. These results demonstrate that `NeCo` is not limited to vision foundation models but can also be effectively applied to vision-language models.

Table 13: In-context scene understanding benchmark (mIoU). Dense nearest neighbor retrieval performance is reported across various training data proportions on ADE20k and Pascal VOC datasets.

| Method | Backbone | Params | ADE20K | | | | Pascal VOC | | | |
|---|---|---|---|---|---|---|---|---|---|---|
| | | | 1/128 | 1/64 | 1/8 | 1/1 | 1/128 | 1/64 | 1/8 | 1/1 |
| CLIP | ViT-B/16 | 85M | 5.8 | 6.5 | 8.7 | 11.3 | 25.3 | 27.8 | 33.4 | 33.9 |
| + NeCo | ViT-B/16 | 85M | **17.7** | **19.8** | **22.9** | **24.2** | **62.8** | **63.5** | **65.1** | **66.2** |
| SigLIP | ViT-B/16 | 85M | 6.0 | 7.1 | 9.0 | 10.6 | 25.3 | 27.8 | 32.2 | 33.9 |
| + NeCo | ViT-B/16 | 85M | **15.9** | **18.4** | **20.7** | **21.9** | **60.9** | **62.0** | **62.5** | **63.1** |

Table 14: **Linear segmentation performance.** A linear segmentation head is trained on top of the frozen spatial features obtained from different feature extractors. We report the mIoU scores achieved on the validation sets of 5 different datasets.

| Method | Backbone | Params | Pascal VOC | ADE20K | COCO-Stuff | COCO-Things | Cityscapes |
|---|---|---|---|---|---|---|---|
| CLIP | ViT-B/16 | 85M | 44.3 | 13.8 | 43.1 | 42.0 | 27.7 |
| +NeCo | ViT-B/16 | 85M | **68.2** | **25.8** | **56.1** | **48.9** | **41.0** |
| SigLIP | ViT-B/16 | 85M | 44.6 | 15.9 | 36.2 | 46.0 | 32.2 |
| +NeCo | ViT-B/16 | 85M | **70.1** | **29.4** | **55.3** | **69.9** | **42.0** |

## B.6 GENERALIZING NECO TO BROADER TASKS AND DATASETS

we report the performance of various backbones, including DINOv2R finetuned with the `NeCo` loss, on out-of-distribution datasets such as Cityscapes (Cordts et al., 2016)(semantic segmentation) and NYUd (Couprie et al., 2013)(monocular depth estimation). As demonstrated in Table 15a and

Table 15c, `NeCo` consistently enhances the generalization of all backbones on the Cityscapes dataset. Furthermore, even for tasks that diverge from semantic segmentation, such as depth estimation, `NeCo` reduces the DINOv2R error by around 2%. These findings highlight that `NeCo` not only maintains but improves the generalization capability of DINOv2R features across diverse tasks. For completeness, we also include results on (Markus Gerke, 2014) in Table 15b, showing that `NeCo` consistently enhances the performance, even on out-of-distribution datasets.

Table 15: **Performance comparison on the Cityscapes, NYUd, and Vaihingen datasets.**

(a) Linear segmentation on Cityscapes

| Pretraining | Original | + `NeCo` |
|---|---|---|
| DINO-B/16 | 41.8 | **42.5** |
| iBot-B/16 | 42.9 | **44.5** |
| Leopard-B/16 | 43.8 | **44.5** |
| DINOv2XR-B/14 | 52.3 | **56.0** |
| DINOv2-B/14 | 51.4 | **57.7** |
| DINOv2R-B/14 | 53.4 | **58.9** |

(b) Linear segmentation on Vaihingen

| Pretraining | Original | + `NeCo` |
|---|---|---|
| CLIP-B/16 | 25.7 | **28.8** |
| SigLIP-B/16 | 26.9 | **28.3** |
| DINOv2R-S/14 | 31.9 | **32.8** |
| DINOv2XR-B/14 | 34.2 | **35.9** |
| Dinov2R-B/14 | 34.9 | **35.3** |

(c) Linear depth prediction on NYUd

| Backbone | Pretraining | RMSE |
|---|---|---|
| | DINOv2 | **0.460** |
| | + `NeCo` | 0.461 |
| ViT-S/14 | DINOv2XR | 0.456 |
| | + `NeCo` | **0.453** |
| | DINOv2R | 0.455 |
| | + `NeCo` | **0.455** |
| | DINOv2 | 0.412 |
| | + `NeCo` | **0.385** |
| ViT-B/14 | DINOv2XR | 0.410 |
| | + `NeCo` | **0.397** |
| | DINOv2R | 0.410 |
| | + `NeCo` | **0.387** |

### B.7 EXTRA ABLATIONS

**Sorting Steepness.** In Table 16e, we vary the sorting steepness, denoted by $\beta$, for both teacher and student networks to evaluate the influence of hard or soft nearest neighbor assignments. The performance improves when the teacher's steepness is higher or equal to the student's, consistent with previous findings (Caron et al., 2021). Our best results are achieved when both networks have equal steepness. However, extreme steepness values (*e.g.*, 1000) harm performance. This is because sorting patch similarities lacks clear boundaries, and formulating it as a hard assignment can force incorrect orderings, negatively impacting performance.

**Training Epochs.** We show the performance across different training epochs in Table 16f. As the table shows, even after just one epoch of training, DINOv2R improves by 1% to 3% across various metrics. The performance continues to increase with more training epochs, but the improvements become smaller after 25 epochs, which is the number used in the paper.

**Patch Similarity Metric.** We show the effect of using different metrics for computing pair-wise patch similarity in Table 16c. As the table shows, Cosine similarity is consistently better than Euclidean distance. We have used cosine similarity in all our experiments.

**ROI-Align Effect.** ROI-Align component can be removed if the multi-crop augmentation is omitted or by using the same Global Crop across the teacher and student branches. We ablate the effect of removing ROI align in Table 16d. As shown by the results, this removal negatively affects performance due to using weaker self-supervised supervision through the augmentations.

Table 16: **Ablations of the key parameters of our method.** We evaluate the models by training a linear layer on top of the frozen representations (Lin.) or using the in-context (IC) evaluation of Balazevic et al. (2023) using the validation images for PascalVOC12 and ADE20k. Note, that the In-Context Learning (IC) is done on the 1/128 fraction of each dataset used.

(a) Nearest-neighbour selection

| NN | Pascal Lin. | Pascal IC | ADE20K Lin. | ADE20K IC |
|---|---|---|---|---|
| intra | 78.1 | 61.2 | 36.3 | 21.3 |
| inter | **78.9** | **62.0** | **37.3** | **21.7** |

(b) Sorting algorithm

| Method | Pascal Lin. | Pascal IC | ADE20K Lin. | ADE20K IC |
|---|---|---|---|---|
| ✗ | 47.3 | 17.8 | 15.8 | 5.1 |
| Odd-even | **78.9** | **62.2** | 37.0 | 21.6 |
| Bitonic | **78.9** | 62.0 | **37.3** | **21.7** |

(c) Patch similarity

| Metric | Pascal Lin. | Pascal IC | ADE20K Lin. | ADE20K IC |
|---|---|---|---|---|
| Euc. | 78.1 | 60.2 | 36.4 | 21.1 |
| Cos. | **78.9** | **62.0** | **37.3** | **21.7** |

(d) ROI Align

| | Pascal Lin. | Pascal IC | ADE20K Lin. | ADE20K IC |
|---|---|---|---|---|
| ✗ | 75.8 | 60.2 | 35.6 | 19.3 |
| ✓ | **78.9** | **62.0** | **37.3** | **21.7** |

(e) Sorting steepness

| (Std, Tch) | Pascal Lin. | Pascal IC | ADE20K Lin. | ADE20K IC |
|---|---|---|---|---|
| (10,100) | 78.3 | 60.5 | 36.2 | 20.8 |
| (1000,100) | 74.5 | 48.3 | 27.8 | 16.5 |
| (1000,1000) | **79.0** | 61.5 | 36.6 | 21.2 |
| (100,100) | 78.9 | **62.0** | **37.3** | **21.7** |

(f) Training Epochs

| Epochs | Pascal Lin. | Pascal IC | ADE20K Lin. | ADE20K IC | Exec. Time |
|---|---|---|---|---|---|
| 1 | 76.8 | 60.5 | 35.8 | 20.1 | 0.5h |
| 2 | 77.7 | 62.7 | 36.6 | 21.4 | 1h |
| 4 | 79.0 | 64.7 | 37.8 | 22.4 | 2h |
| 8 | 80.0 | 65.6 | 38.9 | 22.9 | 5h |
| 16 | 80.6 | 66.1 | 39.5 | 23.3 | 10h |
| 25 | 81.3 | 66.5 | 40.1 | 23.7 | 19h |
| 50 | 81.6 | 66.7 | 40.2 | 23.8 | 40h |

## B.8 COMPUTATIONAL ANALYSIS

We provide a detailed runtime analysis for DINO, CrIBo, TimeT, and `NeCo` in Table 17. All experiments are conducted on 8 NVIDIA RTX A6000-46GB GPUs. The results are reported on COCO-Things linear segmentation. The results show `NeCo` significantly improves computational efficiency and performance. First, DINO and CrIBo are finetuned for 25 additional epochs starting from their existing checkpoints to match the extra training performed by `NeCo`. As the table shows, with the same number of extra epochs, `NeCo` outperforms both models across all metrics, demonstrating that the improvement stems from the proposed loss function rather than extended training. Secondly, with only 2.5 GPU hours of extra training on top of CrIBo, `NeCo` boosts its performance in linear segmentation by 3.7%. These results show that `NeCo` not only enhances computational efficiency but also achieves superior results.

## C ADDITIONAL VISUALIZATIONS

**Visualization of nearest patch retrieval.** In Figure 4, we take one patch from an image in Pascal VOC as the query and retrieve its seven nearest patches across the dataset. We compare `NeCo` against DINOv2R. As illustrated, the nearest neighbors retrieved by `NeCo` are not only more relevant compared to DINOv2R but also more precise, successfully finding nearest patches not only within the same object but also within object parts.

In Figure 5, we show some borderline cases where DINOv2R retrieves more relevant patches than our method. `NeCo` occasionally retrieves patches of similar parts from different objects. For example, a patch from a bicycle wheel might be matched with a motorcycle wheel. Additionally, since we rely on

Table 17: **Computational analysis and segmentation performance.** `NeCo` demonstrates superior computational efficiency, requiring only 2.5 GPU hours to enhance CrIBo's performance by 3.7% in linear segmentation.

| Method | Dataset | Epoch Time (Min:Sec) | Init | Epochs | GPU Hours | K=GT | K=500 | LS |
|--------|---------|---------------------|------|--------|-----------|------|-------|-----|
| DINO | ImageNet | 15:33 | Random | 800 | ∼8 days | 5.4 | 19.2 | 43.9 |
| DINO | ImageNet | 15:33 | Random | 800 + 25 | ∼8 days + 6.5h | 7.0 | 20.3 | 37.4 |
| TimeT | YTVOS | 3:12 | DINO | 30 | ∼8 days + 2h | **18.4** | 44.6 | 58.2 |
| NeCo | COCO | 4:48 | DINO | 25 | ∼8 days + 1.6h | 16.9 | **50.0** | **62.4** |
| CrIBo | ImageNet | 20:37 | Random | 800 | ∼11 days | 14.5 | 48.3 | 64.3 |
| CrIBo | ImageNet | 20:37 | Random | 800 + 25 | ∼11 days + 9h | 15.0 | 48.5 | 64.3 |
| NeCo | COCO | 4:48 | CrIBo | 25 | ∼11 days + 2.5h | **21.1** | **54.0** | **68.0** |

cropping to induce nearest neighbor similarity, small objects in the input, which may not significantly affect the overall semantics, can alter the semantics at the patch level, leading to unexpected nearest neighbors, as seen in the case of the sheep photo.

**Visualization of clustering and Overclustering.** We display the visualizations for both the clustering and overclustering approaches in Figure 7 and Figure 6, respectively. For the clustering approach, detailed in Table 1b, we apply cluster-based foreground extraction combined with community detection to identify the foreground regions from features extracted across the entire dataset. The extracted features are then masked with the extracted masks and clustered according to the number of objects in the dataset, which is 21 for Pascal. As shown in Figure 7, this process successfully assigns unique cluster IDs to the detected objects and accurately sketches their boundaries.

For overclustering, we don't extract foreground regions and instead cluster all the features into a significantly higher number of clusters compared to the ground truth. For Pascal, this number ($K$) is set to 100. Figure 6 illustrates the results. We observe a similar effect as with clustering, except that some objects, such as humans or certain animals, are partitioned into their constituent parts, which remain relatively consistent across different samples.

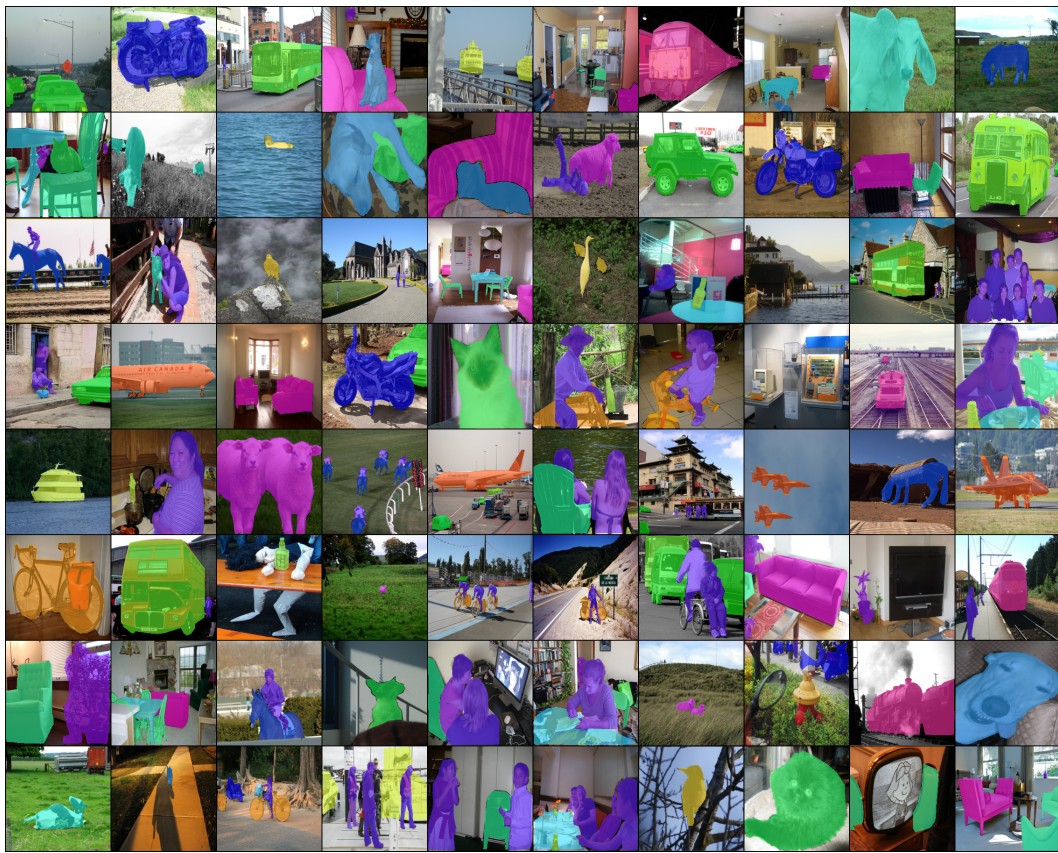

Figure 3: **Pascal VOC visualizations.** We overlay the ground truth on top of a subset of images in Pascal VOC. These images and their ground truth segmentation maps are used for our tasks, such as visual in-context learning and linear segmentation.

(a) DINOv2R

(b) NeCo

Figure 4: **Nearest patch retrieval.** Comparison of nearest neighbor retrieval results between NeCo and DINOv2R on Pascal VOC. For each query patch, NeCo retrieves more relevant and precise nearest patches, accurately identifying patches within the same object and object parts.

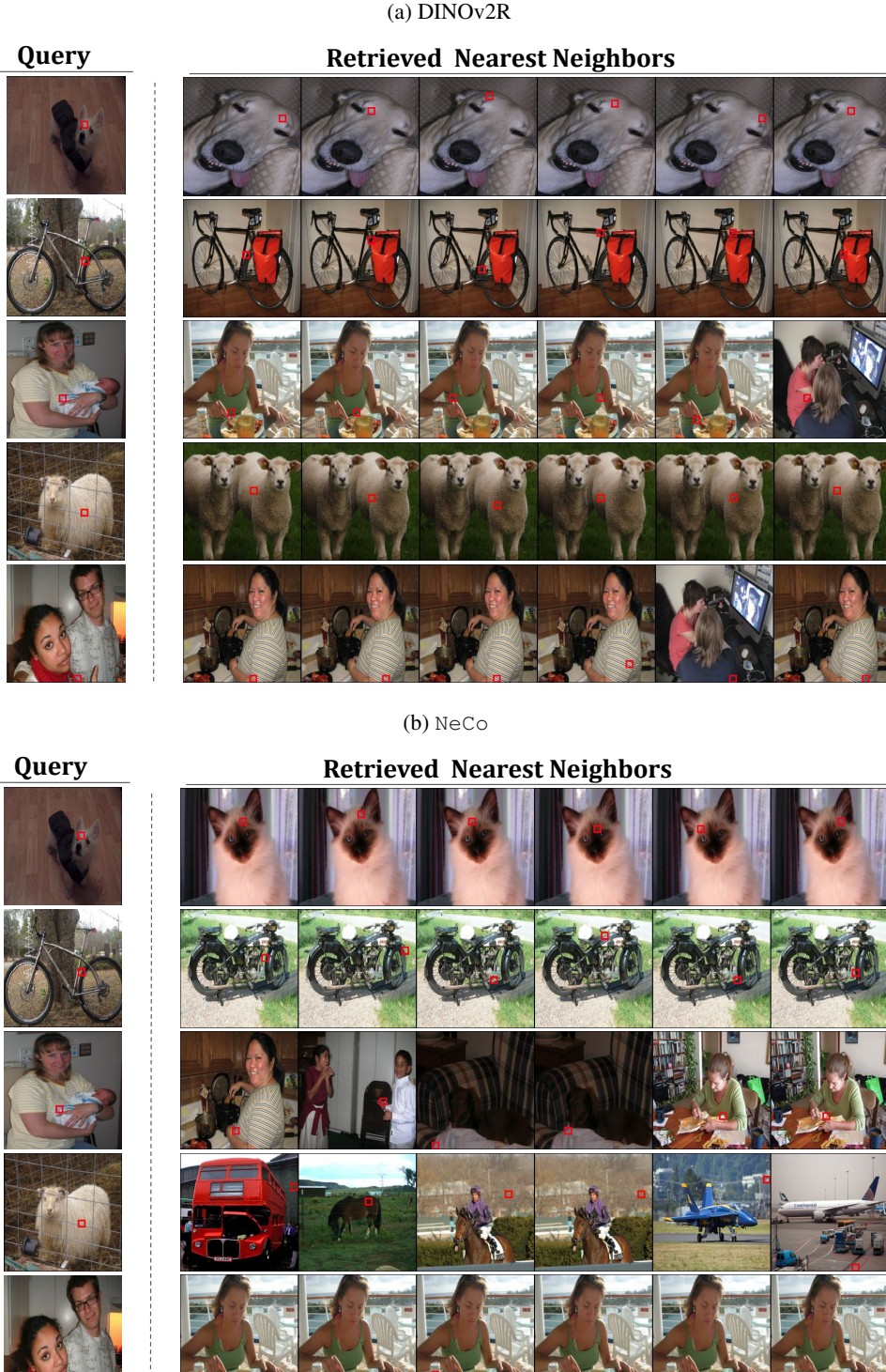

Figure 5: **Borderline cases.** NeCo , sometimes retrieves patches of similar parts from different objects. For example, a patch from a bicycle wheel might be matched with a motorcycle wheel. Additionally, since we rely on cropping to induce nearest neighbor similarity, small objects in the input, which may not significantly affect the overall semantics, can alter the semantics at the patch level, leading to unexpected nearest neighbors, as seen in the case of the sheep photo.

(a) Input

(b) DINOv2R

(c) NeCo

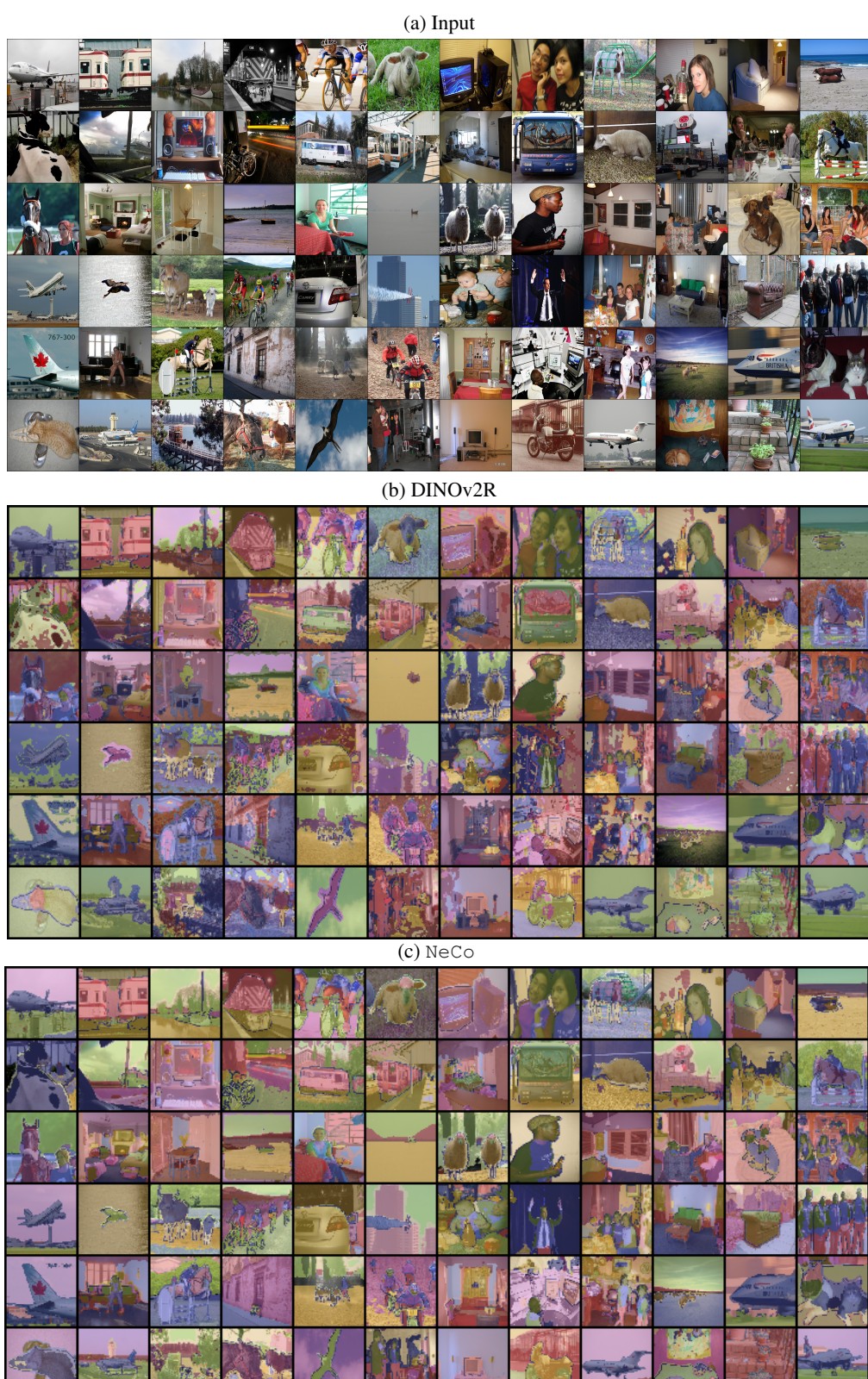

Figure 6: **DINOv2R and NeCo overclustering visualizations on Pascal for K=100.** NeCo localizes objects more precisely with tighter boundaries.

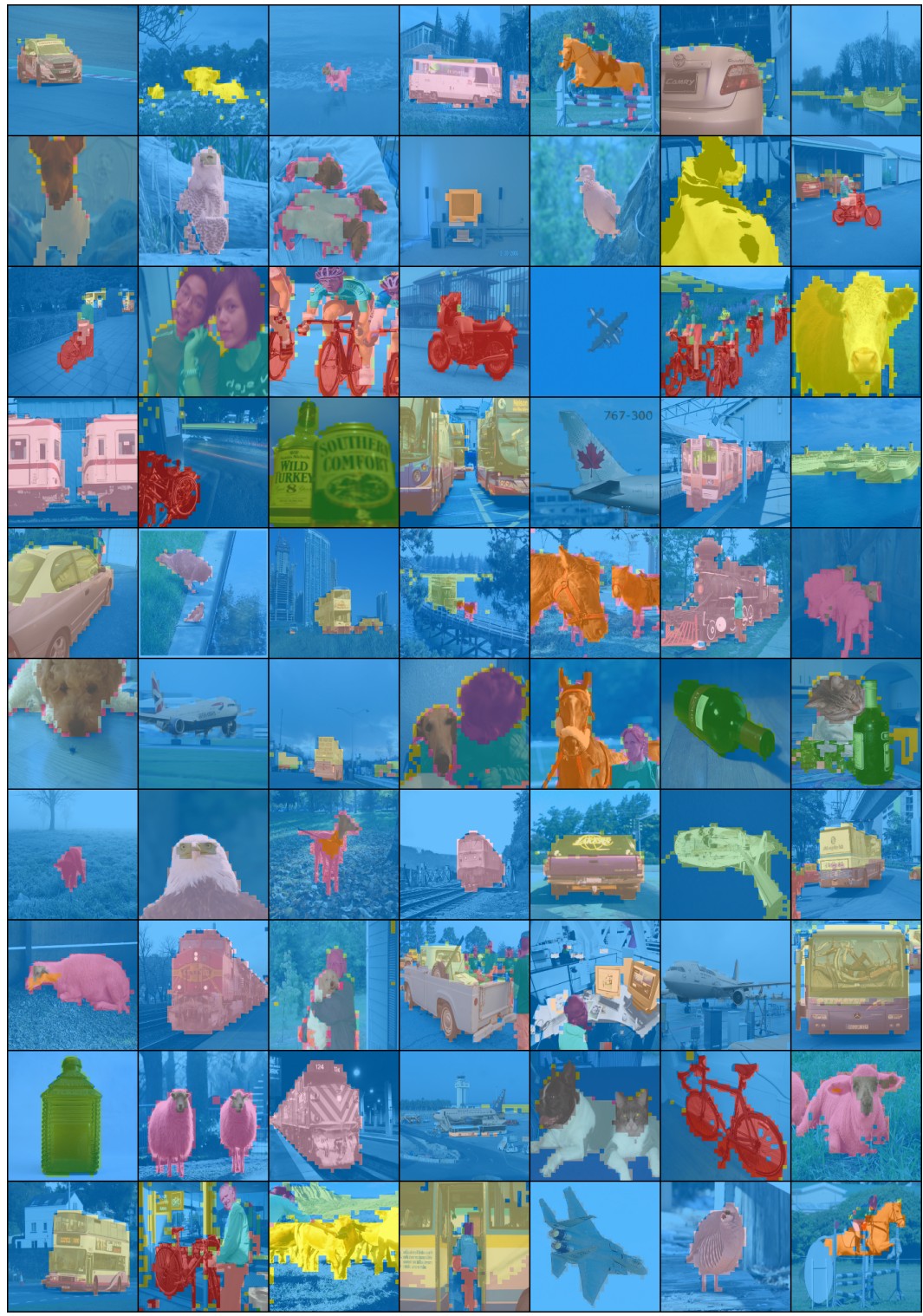

Figure 7: **Fully unsupervised segmentation on Pascal for K=21.** We extract foreground masks using the CBFE+CD method, following the approach outlined in Ziegler & Asano (2022). These masks are then clustered into the number of objects present in the Pascal dataset, with $K = 21$. As demonstrated, NeCo yields distinct and accurate segmentation maps for each object.

# D DATASET DETAILS

**Pascal VOC 2012** (Everingham et al.) This dataset, the latest split version of trainaug, features 10,582 images and their annotations distributed across 21 classes, with one referring to the background class. The validation set consists of 1,449 images. Following Van Gansbeke et al. (2021) we ignore unlabelled objects as well as the boundary class. Moreover, for hyper-parameter tuning of the fully unsupervised segmentation method (Ziegler & Asano, 2022) that we apply on our method, we use the PVOC12 train split with 1464 images. Figure 3 shows the dataset images overlaid by the annotations.

**Pascal Context** (Everingham et al., 2010) This scene-centric dataset includes 4,998 training images covering 60 semantic classes, including the background. The validation set consists of 5,105 images. We use this dataset for the Linear Segmentation and Segmenter experiments, via the MMSegmentation Library (MMSegmentation Contributors, 2020).

**COCO-Stuff 164K** (Caesar et al., 2018) This scene-understanding dataset includes labels across 91 "stuff" categories and 80 "things" categories. The training set comprises 118,000 images, and the validation set contains 5,000 images. We follow the same setup as Ziegler & Asano (2022) and thus we use the COCO benchmark in two ways to isolate further the given object definitions.

Concisely, we begin by extracting stuff annotations, which refer to objects without clear boundaries and often found in the background, using the COCO-Stuff annotations (Caesar et al., 2018). Then, we consolidate the 91 detailed labels into 15 broader labels, as described in Ji et al. (2019) and we assign the general label "other" to non-stuff objects, as this label lacks specific semantic meaning. Non-Stuff objects are ignored during training and evaluation. We indicate this version of the dataset within our work as COCO-Stuff used in Overclusterring and Linear Segmentation in Appendix A.2.

Next, we extract foreground annotations utilizing the panoptic labels from Kirillov et al. (2019). We combine the instance-level annotations into object categories using a script provided by the authors. Additionally, we consolidate the 80 detailed categories into 12 broad object classes.The background class is ignored during training and evaluation. This leads as to the COCO-Thing version of the dataset which we use for the Overclusterring and our Linear Segmentation in Appendix A.2.

**ADE20K** (Zhou et al., 2017) The dataset is a collection of images used for semantic segmentation tasks, featuring finely detailed labels across 150 unique semantic categories. Some of the categories include stuffs like sky and grass, as well as distinguishable objects like person, and a car. Overall, it includes a wide variety of scenes, with 20,210 images in the training set and 2,000 images in the validation set, making it one of the most challenging and diverse datasets for scene understanding. We use the full dataset in our experiments. In our experiments, we ignore the *others* label of the dataset.

**Imagenet** (Russakovsky et al., 2015) The dataset, is a large-scale visual database designed for use in visual object recognition research. It contains over 1.3 million images categorized into 1,000 object classes. Each image is labeled with detailed annotations, making it a critical resource for training and evaluating machine learning models, particularly in the field of computer vision. In our work, we also explore training on part of the Imagenet, the Imagenet100k that consists 100K images across 100 classes, from the original dataset.

**SPair-71k** (Min et al., 2019) is a large-scale dataset of image pairs explicitly designed for semantic correspondence. It was constructed from images extracted from the well-known datasets of PASCAL (Everingham et al., 2010; Xiang et al., 2014). The task involves establishing correspondences at the pixel level between instances of objects in different images of the same class. There are 18 object categories, diversified between rigid and non-rigid objects, such as cars, aeroplanes, cats, and humans. Of these, 8 categories represent non-rigid objects, particularly challenging for semantic matching due to their deformability: cats, cows, humans, among others.

Each image is paired with its corresponding class-specific keypoints to help determine salient object parts. Furthermore, the pairs of images were annotated with a measure of viewpoint variation, describing how far the perspective can change between two images of the same category of an object. This annotation is done by human annotators for high-quality evaluation data. The combination of rich variability in object pose, appearance, and occlusion with background clutter, together with dense high-quality ground-truth correspondences, renders SPair-71k a uniquely challenging but comprehensive dataset to advance the frontiers of semantic correspondence and object matching.

