# OpenReview forum: "Near, far: Patch-ordering enhances vision foundation models' scene understanding"
_ICLR.cc/2025/Conference — ICLR 2025 Poster_

### Official Review · Reviewer_KF4P · 2024-10-31

**Soundness:** 3
**Presentation:** 3
**Contribution:** 3
**Rating:** 8
**Confidence:** 3

**Summary:**

The paper introduces NeCo, a self-supervised training loss that enforces patch-level nearest neighbor consistency between a student and teacher model. Unlike binary contrastive methods, NeCo leverages fine-grained learning signals by sorting spatially dense features relative to reference patches. By applying differentiable sorting on top of pretrained representations like DINOv2, NeCo achieves superior results across various models and datasets with minimal GPU time.

**Strengths:**

1. It makes sense to perform self-supervised learning by bringing Neighbor Patch Consistency closer.

2. Although the overall method design is based on existing simple strategies, it shows significant performance improvements. The proposed method achieves SOTA on several downstream tasks.

3. Further self-supervised training based on existing pretrained models (e.g., DINOv2) can significantly reduce computational cost and GPU training time.

**Weaknesses:**

1.The comparative experiments in the paper are not comprehensive. If DINOv2 is used as a baseline, it is necessary to align with the downstream tasks validated in the DINOv2 paper as much as possible. Additionally, if the goal is to claim dense prediction capability, it is also important to validate the model’s representation abilities on more instance and dense recognition tasks, such as CityScapes, KITTI, and NYUd.

2.Regarding pairwise distance computation, if this method only calculates distances within a single batch, it results in a limited computational space and may not capture all potential patches that are near the selected patches.

3.Obtaining patch features prior to distance computation heavily relies on the feature extraction capability of the pretrained foundation model. If the pretrained foundation model introduces noise in feature extraction (a common phenomenon), how does your method prevent the accumulation of such errors?

4.The use of cosine similarity for distance calculation may not be the most accurate choice, as it lacks an ablation study.

5.Does the proposed Patch Neighbor Consistency self-supervised strategy have generalizability? For instance, can it be applied to more foundation models like CLIP, SIGLIP, SAM, etc.?  In other words, can the proposed method adapt to more pretrained models as in the experiment in Table 3?

6.Additionally, is this method suitable for transformer models trained from scratch, assuming it can use another foundation model to extract features for distance computation?

**Questions:**

Please refer to the weaknesses. I also have an additional empirical question: Would incorporating a 'repel' signal for distant patches (similar to contrastive loss) alongside the computation of near patch consistency enhance the performance of the proposed method?

---

> ### Author Response · Authors · 2024-11-22
>
> **More instance and dense-level tasks**
> ### *More Dense tasks:*
>
> We appreciate the reviewer’s concern and have added linear segmentation
> performance on CityScapes and NYUd dataset to our linear segmentation experiments, as shown by Table. 2. Here are the results :
>
>  *Table 1. Linear segmentation on Cityscapes (mIoU)*
>
> | Backbone    | Original | + NeCo   |
> |-------------|----------|----------|
> | Dino        | 40.7     | **43.7** |
> | iBot        | 43.0     | **44.8** |
> | TimeT       | 42.2     | **42.8** |
> | Leopart     | 44.5     | **45.5** |
> | Dinov2R     | 48.9     | **52.9** |
>
> ---
>
>  *Table 2. Linear depth prediction on NYUd (Root-mean-squared-error, lower is better)*
>
> #### **Backbone (ViT-S)**
>
> | Backbone    | RMSE     |
> |-------------|----------|
> | Dinov2      | 0.460    |
> | Dinov2R     | 0.456    |
> | Dinov2R+NeCo | **0.453** |
>
> #### **Backbone (ViT-B)**
>
> | Backbone    | RMSE     |
> |-------------|----------|
> | Dinov2      | 0.412    |
> | Dinov2R     | 0.410    |
> | Dinov2R+NeCo | **0.397** |
>
> As demonstrated in the results, the features learned by NeCo consistently improve DINOv2R on different dense tasks and datasets, which support the generality of our method.
>
> For the KITTI dataset, we expect results similar to those observed on Cityscapes. However, due to the large image sizes in KITTI, experiments take longer, and we plan to include the results in the final version.
>
> ### *Instance-level tasks:*
> The table below shows the performance of DINO and DINO fine-tuned with NeCo for object detection on the COCO dataset, evaluated using the VitDet benchmark with a ViT-small model. Our method improves the DINO backbone, increasing the validation Box AP slightly from 42.9 to 43.0 and the Mask AP from 38.6 to 38.7.
>
>  *Table 2. Object detection performance based on VitDet benchmark*
>
> | Backbone    | Epochs | Val Box AP | Val Mask AP |
> |-------------|--------|------------|-------------|
> | Dino        | 12     | 42.9       | 38.6        |
> | Dino+NeCo | 12     | **43.0**   | **38.7**    |
>
>
> **Could single-batch distance computation miss some semantically close patches?**
>
> We agree with the reviewer that single-batch distance computation may not capture all pairwise relationships between patches. We have done an ablation study on the impact of batch size on performance, as presented in Table 6.e. For clarity, we include the table here for reference.
>
> *Paper table 6. e: Ablation on batchsize*
>
> | Batch Size | PVOC LS | PVOC IC | ADE20K LS | ADE20K IC |
> |------------|----------|----------|-----------|-----------|
> | 4          | 76.2     | 60.2     | 35.6      | 20.2      |
> | 8          | 76.8     | 61.1     | 36.3      | 20.8      |
> | 16         | 77.7     | 60.9     | 36.7      | 21.1      |
> | 32         | 78.2     | 61.4     | 37.1      | 21.4      |
> | 64         | **78.9** | **62.0** | **37.3**  | **21.7**  |
>
>
> As shown, performance improves over the baseline even with smaller batch sizes, indicating that the most useful patches are likely already present in the image itself and not in the reference patches. However, we do observe gains with increased batch sizes, likely due to the increased presence of further semantically close patches.
>
> **How does the method handle scenarios where the foundation model produces noisy features?**
>
> As noted by the reviewer, the quality of the initial features allows our method to start with a strong pairwise distance matrix, which is further refined during training. We conducted an extensive ablation study on the effect of pretraining, as shown in Table 3. This includes results from five additional pretraining methods beyond DINOv2R, demonstrating that even less robust models serve as suitable starting points for our method.
>
> **Ablate on different similarity measures other than cosine**
>
> As requested by the reviewer we have added an ablation study on the distance metric used to create similarity metrics. As the table shows, Cosine similarity is consistently better than Euclidean distance. We have used cosine similarity in all our experiments.
>
>
> *Table 3. Linear segmentation (LS) and vision in-context performance (IC) on top of the frozen backbone*
> | Method | PVOC LS | PVOC IC | ADE20k LS | ADE20k IC |
> |--------|---------|---------|-----------|-----------|
> | Euc    | 78.1    | 60.2    | 36.4      | 21.1      |
> | Cos    | **78.9** | **62.0** | **37.3**  | **21.7**  |
>
> **Add clip, sigclip or sam to your tables**
>
> We present the results for CLIP and SigLIP on linear segmentation and visual in-context learning, as requested by the reviewer.
>
> --- continued ---

---

> ### Author Response · Authors · 2024-11-22
>
> *Table 4. Linear segmentation (mIoU)*
>
> | Backbone (vitb) | PVOC  | ADE20K | COCO-Stuff | COCO-Thing | Cityscapes |
> |-----------------|-------|--------|------------|------------|------------|
> | CLIP            | 44.3  | 13.8   | 43.1       | 42.0       | 27.7       |
> | CLIP+NeCo       | **68.2** | **25.8** | **56.1**   | **48.9**   | **41.0**   |
> | SIGLIP          | 44.6  | 15.9   | 36.2       | 46.0       | 32.2       |
> | SIGLIP+NeCo     | **70.1** | **29.4** | **55.3**   | **69.9**   | **42.0**   |
>
>
> *Table 5. Pascal VOC - Vision In Context Learning with NN Dense Retrieval (mIoU)*
>
> | Backbone (ViT-B) | 1/128 | 1/64 | 1/8 | full |
> |------------------|-------|------|-----|------|
> | CLIP             | 25.3  | 27.8 | 33.4 | 33.9 |
> | CLIP+NeCo        | **62.8** | **63.5** | **65.1** | **66.2** |
> | SIGLIP           | 25.3  | 27.8 | 32.2 | 33.9 |
> | SIGLIP+NeCo      | **60.9** | **62.0** | **62.5** | **63.1** |
>
>
> *Table 6. ADE20k - Vision In Context Learning with NN Dense Retrieval (mIoU)*
>
> | Backbone (ViT-B) | 1/128 | 1/64 | 1/8 | full |
> |------------------|-------|------|-----|------|
> | CLIP             | 5.8   | 6.5  | 8.7 | 11.3 |
> | CLIP+NeCo        | **17.7** | **19.8** | **22.9** | **24.2** |
> | SIGLIP           | 6.0   | 7.1  | 9.0 | 10.6 |
> | SIGLIP+NeCo      | **15.9** | **18.4** | **20.7** | **21.9** |
>
> As shown in the tables above, NeCo improves the performance of both CLIP and SigLIP by approximately 12% to 37% across various benchmarks. These results demonstrate that NeCo is not limited to vision foundation models but can also be effectively applied to vision-language models.
>
> **Can it be trained from scratch, assuming that the similarities are given from another model?**
>
> Thank you for the interesting suggestion. We think this could be explored in future research, as this type of model distillation  would be different from the main focus of our work. Our approach begins with a pretrained teacher network and improves its performance through further fine-tuning. In contrast, the model trained with the reviewer’s proposed method would likely be limited by the teacher/distilling network.
>
> **Would incorporating a repel signal similar to contrastive loss be helpful for the training?**
>
> The goal of this paper is to develop a feature space in which, for a given input, patches representing the same object exhibit similar features, while patches from different objects show distinct features. To effectively model this, we need a hierarchy of similarities. Contrastive methods, however, potentially violate this hierarchy by only aiming to bring similar patches together while pushing dissimilar ones apart, without explicitly enforcing the nuanced ordering that we require. In contrast, our sorting-based approach maintains this hierarchy by explicitly enforcing the relative similarity of patches within a more structured ordering. Moreover, sorting also generates repelling signals, as it brings similar patches together while simultaneously distancing them from dissimilar patches; yet, the repelling effect in sorting is milder than in contrastive methods.
>
> We thank the reviewer once again for the valuable feedback. We hope the response adequately addresses the concerns raised and encourages the reviewer to improve the score.

---

> > ### Comment · Reviewer_KF4P · 2024-11-25
> > **Responses to Author's Rebuttal**
> >
> > I believe that the instance- and dense-level tasks sufficiently demonstrate the effectiveness of your proposed method. Additionally, conducting experiments with more large models as base models addresses my concern and further proves the generalizability of your approach. I find this to be a thorough and well-prepared rebuttal response, and I am inclined to increase my rating score.
> >
> > However, I think the experiments in Table 6 of the paper do not directly prove that "calculating distances within a single batch" is the optimal design for the following reasons:
> > 1) Increasing the batch size requires simultaneous adjustment of the learning rate.
> > 2) Additional experiment (i.e., batchsize = 1) should be conducted to verify the claim that "useful patches are likely already present in the image itself."

---

> ### Author Response · Authors · 2024-11-25
>
> Dear Reviewer KF4P,
>
> Thank you again for the time and effort spent on your thorough review of our paper. Since the author-reviewer discussion deadline is fast approaching, we kindly ask for feedback on our responses. We would be happy to discuss more if there are still some open questions.
>
> Best Regards,
>
> Authors

---

> ### Author Response · Authors · 2024-11-27
>
> **Increasing the batch size requires simultaneous adjustment of the learning rate:**
>
> We agree with the reviewer. In our work, we employed the ADAM optimizer, which dynamically adjusts the learning rate based on training dynamics. As a result, we expect that the adjustments in learning rate for different batch sizes would lead to a similar overall impact on the training process.
>
> **Additional experiment (i.e., batchsize = 1) should be conducted to verify the claim that "useful patches are likely already present in the image itself.":**
>
> As requested by the reviewer, we conducted an ablation study with a batch size of 1. Training with this configuration takes longer because processing each sample individually reduces the efficiency of computations. Consequently, the model was trained for only one epoch for this response. The results are presented in the table below:
>
>
> ***Table 1. Evaluating the performance gain for batchsize=1***
>
> | Batch Size      | PVOC LS | PVOC IC | ADE20K LS | ADE20K IC |
> |-----------------|---------------|-------------|---------------|-------------|
> | DINOv2R        | 74.2          | 53.0        | 35.0          | 19.6        |
> | + NeCo (bs=1)  | **76.0**          | **60.0**        | **35.4**          | **20.1**        |
>
> As shown, even with a batch size of 1 and just one epoch of training, NeCo demonstrates improvements in DINOv2R performance ranging from 0.4% to 7% across different metrics and datasets. Compared to our best results, approximately 30% of the linear evaluation gains and 80% of the in-context gains on Pascal VOC can be achieved using a single image in each batch. Similarly, for ADE20K, about 30% of the gains can also be achieved using patch comparisons within the image alone. Moreover, training with a batch size of 1 actually allows one to have a very small memory footprint of 3517MB during training, allowing us to actually improve the DINOv2R model with a single GTX 1080, a GPU from 2016. We find this surprising and interesting; thanks for the suggestion!
>
> For further insights into the effect of patch selection, we had already included an additional ablation study in Table 10A (Appendix) and lines 505–509, which examines the impact of inter-image selection (ie image patches are compared to both patched within the same image and across patches from the references in the batch) and intra-image patch selection (ie only within the image) on performance. We add this table for your convenience below :
>
> ***Table 10.e Nearest neighbor selection***
>
> | NN       | PVOC LS | PVOC IC | ADE20K LS | ADE20K IC |
> |----------|---------|---------|-----------|-----------|
> | Intra    | 78.1    | 61.2    | 36.3      | 21.3      |
> | Inter    | 78.9    | 62.0    | 37.3      | 21.7      |
>
> As shown, the results for both are very close, with most of the gains achievable through intra-image patch selection, further supporting our claims.

---

### Official Review · Reviewer_1JH6 · 2024-11-01

**Soundness:** 3
**Presentation:** 4
**Contribution:** 3
**Rating:** 6
**Confidence:** 4

**Summary:**

The paper introduces NeCo, a self-supervised training algorithm that enhances patch-level nearest neighbor consistency across student and teacher models, aiming at improving scene understanding in vision foundation models. NeCo leverages differentiable sorting on pretrained representations, such as DINOv2, to boost learning signals and performance across various models and datasets. The method requires minimal training time and establishes new state-of-the-art results in non-parametric in-context semantic segmentation and 3D understanding.

**Strengths:**

1.	The paper is well-written.
2.	Experiments are extensive and convincing, providing rich results on various settings.
3.	The proposed method is simple, interesting, and effective, surpassing previous works by a large margin.

**Weaknesses:**

1.	The major contribution of NeCo seems to be introducing sorting consistency into visual representation pretraining. In pages 4-5, the description of differentiable swapping and sorting takes majority of space in Section Method, but these contents are proposed by previous works and simply pasted here.
2.	The method requires a pretrained backbone (e.g. DINO) for visual representation learning, and outperforms previous methods. However, some previous learning methods (e.g. MAE) does not require a pretrained backbone, so the improvement may partially come from the additional information brought by the pretrained backbone DINO. Is it possible to use the sorting consistency between different views to learning representation without the pretrained backbone teacher?

**Questions:**

1.	Please give more details on sampling and collecting patch features with ROIAlign. How the box coordinates of patches are sampled? Also, how the different views of one single image are sampled?
2.	How the sorting algorithm is implemented? Are O(nlogn) algorithms such as quick sorting (other than odd-even sorting mentioned in the manuscript) compatible with differentiable swapping?

---

> ### Author Response · Authors · 2024-11-22
>
> **NeCo's primary contribution is introducing sorting consistency in visual representation pretraining, though much of the Method section on differentiable swapping and sorting relies on prior work.**
>
> The reviewer is correct. This is to make the paper more self-contained and more straightforward for readers unfamiliar with those other works. We aim to make the prior work section even more compact.
>
> **Is the improvement coming from starting from a pre-trained network? Can the method be trained from scratch?**
>
> Most likely, yes, but training Vision Transformers (ViTs) from scratch requires many optimization tricks [1] which also significantly increases training time. Our main contribution lies in demonstrating that, with as little as 0.2% additional compute compared to DINOv2R, we can achieve substantial improvements in dense representation quality. This highlights the efficiency and practicality of our approach in contrast to the resource-intensive process of training ViTs from scratch.
>
> **improvement may partially come from the additional information brought by the pretrained backbone DINO.**
>
> Indeed, this is why we compare against the original pretrained models in all our experiments. Moreover, we also find that the gain does not solely come from using the COCO dataset either: From our experiment of training NeCo on PascalVOC, as shown in Table 6.d, we observe that the resulting performance—77.9 and 36.4 for linear segmentation on Pascal and ADE20K, respectively—exceeds that of the original model, which achieved 74.2 and 35.0, despite DINOv2's training dataset being constructed with PascalVOC.  This improvement demonstrates that the proposed loss function imposes meaningful constraints on the dense features, enabling the model to better utilize the training samples, despite their prior exposure to the DINOv2R loss.
>
> **More details on sampling and collecting patch features with ROIAlign. How are different views sampled?(Question1)**
>
> *Generating the Crops* : We generate multiple views of an image by applying random cropping and resizing using the RandomResizedCrop method from PyTorch's torchvision.transforms module. For global crops, we use a narrower range of scaling parameters s=(0.25, 1), ensuring that larger regions of the image are captured. These global crops are resized to 224×224 resolution. For local crops, which focus on smaller regions, we use scaling parameters s=(0.05,0.25) and resize these crops to 96×96 resolution. Both types of crops are further augmented using horizontal flips, color distortions, and Gaussian blur to increase diversity.  We follow this multi-crop strategy from previous works such as [3, 4].
>
> *Feature Extraction with ROI Align* : For each pair of crops, we compute and store the intersection bounding boxes of their overlapping regions during the cropping process. During feature extraction, these bounding boxes are used with the ROI Align method to sample the corresponding regions from the feature maps. Since features from different crops may vary in size due to differing scales and resolutions, ROI Align resizes these regions to a fixed size (e.g., 7×7) using bilinear interpolation, ensuring spatial alignment and uniform dimensions.
>
> We will make sure to add this explanation to our appendix.
>
> **How the sorting algorithm is implemented? Can we use nlog(n) algorithms?(Question2)**
>
> We have tested different sorting algorithms, as shown in Table 10.b (Appendix). Our method works well with all the algorithms, but bitonic sorting is slightly faster and more efficient than odd-even sorting, so we chose it as the default. Although we haven't tested our method with faster differentiable sorting algorithms, such as  [2] with O(nlog⁡n) complexity, the modular design makes it easy to switch to a faster algorithm if needed, as long as it doesn't negatively affect performance.
>
>
> We thank the reviewer once again for the valuable feedback. We hope the response adequately addresses the concerns raised and encourages the reviewer to improve the score.
>
> [1] - Touvron, et al. "Deit iii: Revenge of the ViT." ECCV 2022.
>
> [2] - Blondel et al. "Fast differentiable sorting and ranking." ICML 2020
>
> [3] - Caron et al. "Emerging properties in self-supervised vision transformers." ICCV. 2021.
>
> [4] - Ziegler et al. "Self-supervised learning of object parts for semantic segmentation." CVPR 2022.

---

> ### Author Response · Authors · 2024-11-25
>
> Dear Reviewer 1JH6,
>
> Thank you again for the time and effort spent on your thorough review of our paper. Since the author-reviewer discussion deadline is fast approaching, we kindly ask for feedback on our responses. We would be happy to discuss more if there are still some open questions.
>
> Best Regards,
>
> Authors

---

> ### Author Response · Authors · 2024-11-30
>
> Dear Reviewer 1JH6,
>
> we have aimed to thoroughly address your comments. If you have any further concerns, we would be most grateful if you could bring them to our attention, and we would be pleased to discuss them.
>
> Sincerely,
>
> Authors

---

### Official Review · Reviewer_hnaH · 2024-11-03

**Soundness:** 2
**Presentation:** 2
**Contribution:** 2
**Rating:** 6
**Confidence:** 3

**Summary:**

The paper introduces a new self-supervised training loss called Patch Neighbor Consistency (NeCo). The key benefit of the proposed method is that the loss yields richer/fine-grained supervision over contrastive approaches. The proposed training scheme is applied over DINOv2 pretrained backbones and shows further improvements on a wide range of downstream tasks.

**Strengths:**

1. The paper is well written, with a wide range of ablations substantiating most of the design choices introduced in the method.
2. The improvements obtained by the proposed method in linear probing and multiview feature consistency evaluations indicate the superior quality of the learned features. Overall the method shows good improvement in a wide range of downstream evaluations.

**Weaknesses:**

1. The main weakness of the proposed method is the baselines. Since the proposed method is introduced as post-training adaptation scheme. It would be fair to continually pretrain DINOv2 and other pretraining methods also by the same duration as NeCo. An ablation on matching training flops/epochs between all the pretraining methods and NeCo would be good to have in the paper.
2. The method has carefully crafted hyperparameters to ensure some overlap between teacher and student patches. Is it possible that the method can do away with ROIAlign by not performing any geometric transformations in data augmentation. At the moment it seems like ROIAlign is used to undo the geometric transformation applied to the image presented to the teacher model.

**Questions:**

It would be good to clarify the main weakness of proposed baselines. An understanding of how the method performs against baseline approaches, when their training flops/epochs are matched can bolster the claims made in the paper.

---

> ### Author Response · Authors · 2024-11-22
>
> **An ablation on matching training flops/epochs between all the pretraining methods and NeCo, including DINOV2R:**
>
> Although we fine-tune DINOv2R for 25 additional epochs, it is important to note that this model, which serves as the main model for NeCo, undergoes these epochs on COCO—a dataset that constitutes only 0.11% of DINOv2R's original training dataset of 142M images. In effect, the 25 training epochs equate to roughly 2% of a full epoch and thus merely **0.2%** of DINOv2R's extensive pretraining (10 epoch pretraining). Despite this minimal training, NeCo improves DINOv2R's performance by at least 6% across several evaluation benchmarks.
> We also acknowledge the reviewer’s concern and have provided a detailed computational analysis in Table 11 (Appendix). Following previous methods [4], we report the total training and epoch time. We will move this table from the appendix to the main paper for visibility. The table has been extended with the results below, and epoch time is adjusted based on dataset size.
>
>  *Table 1. Computational efficiency analysis and evaluation on frozen clustering with K=ground truth or 500 and linear segmentation*
> | Method | Dataset   | Epoch Time | Init   | Epochs    | GPU hours          | K=GT  | K=500 | LS    |
> |--------|-----------|------------|--------|-----------|--------------------|-------|-------|-------|
> | DINO   | ImageNet  | 15:33      | Random | 800       | ~8 days            | 5.4   | 19.2  | 43.9  |
> | DINO   | ImageNet  | 15:33      | Random | 800 + 25  | ~8 days + 6.5h     | 7.0   | 20.3  | 37.4  |
> | TimeT  | YTVOS     | 3:12       | DINO   | 30        | ~8 days + 2h       | **18.4**  | 44.6  | 58.2  |
> | NeCo   | COCO      | 4:48       | DINO | 25       | ~8 days + 1.6h     | 16.9  | **50.0**  | **62.4**  |
> |--------|-----------|------------|--------|-----------|--------------------|-------|-------|-------|
> | CrIBo  | ImageNet  | 20:37      | Random | 800       | ~11 days           | 14.5  | 48.3  | 64.3  |
> | CrIBo  | ImageNet  | 20:37      | Random | 800 + 25  | ~11 days + 9h      | 15.0  | 48.5  | 64.3  |
> | NeCo   | COCO      | 4:48       | CrIBo | 25       | ~11 days + 2.5h    | **21.1**  | **54.0**  | **68.0**  |
>
>
>
> The results, reported on COCO-Things linear segmentation, highlight NeCo's significant improvement in computational efficiency and performance. First, DINO and CrIBo are finetuned for 25 additional epochs starting from their existing checkpoints to match the extra training performed by NeCo.  As the table shows, with the same number of extra epochs, NeCo outperforms both models across all metrics, demonstrating that the improvement stems from the proposed loss function rather than extended training. Secondly, with only 2.5 GPU hours of extra training on top of CrIBo, NeCo boosts its performance in linear segmentation by 3.7%. These results show that NeCo not only enhances computational efficiency but also achieves superior results.
>
> **The method has carefully crafted hyperparameters to ensure some overlap between teacher and student patches**
>
> The augmentation procedure in our work adheres to the standard multi-crop strategy, as established in prior works such as SwAV, DINO, Leopart [1, 2, 3].  This approach generates both global and local views of the image: global crops capture broader context by covering larger portions of the image, while local crops focus on smaller regions to extract fine-grained details. As our loss is dense, we thus require some matching between the corresponding patches. For this, we simply follow [3], to ensure a positive  intersection between global crops and align the views by ROIAlign.
>
> **Is it possible to remove the ROIAlign?**
>
> Yes,  If multi-crop augmentation is omitted, ROI Align can also be removed by using the same Global Crop across the teacher and student branches. However, this removal could negatively affect performance due to using weaker self-supervised supervision through the augmentations. We have run this as an interesting ablation study and the results are reported in the table below:
>
>  *Table 2. Ablating the effect of ROI Align operator*
> | Method        | PVOC LS | PVOC IC | ADE20k LS | ADE20k IC |
> |---------------|---------|---------|-----------|-----------|
> | Without ROI   | 75.8    | 60.2    | 35.6      | 19.3      |
> | With ROI      | **78.9**| **62.0**| **37.3**  | **21.7**  |
>
> We thank the reviewer once again for the valuable feedback. We hope the response adequately addresses the concerns raised and encourages the reviewer to improve the score.
>
> [1] - Caron et al. "Unsupervised learning of visual features by contrasting cluster assignments." NeurIPS 2020
>
> [2] - Caron et al. "Emerging properties in self-supervised vision transformers." ICCV. 2021.
>
> [3] - Ziegler et al. "Self-supervised learning of object parts for semantic segmentation." CVPR 2022.
>
> [4] - Lebailly et al. "CrIBo: Self-Supervised Learning via Cross-Image Object-Level Bootstrapping." ICLR 2024.

---

> ### Author Response · Authors · 2024-11-25
>
> Dear Reviewer hnaH,
>
> Thank you again for the time and effort spent on your thorough review of our paper. Since the author-reviewer discussion deadline is fast approaching, we kindly ask for feedback on our responses. We would be happy to discuss more if there are still some open questions.
>
> Best Regards,
>
> Authors

---

> ### Author Response · Authors · 2024-11-30
>
> Dear Reviewer hnaH,
>
> we have aimed to thoroughly address your comments. If you have any further concerns, we would be most grateful if you could bring them to our attention, and we would be pleased to discuss them.
>
> Sincerely,
>
> Authors

---

> > ### Comment · Reviewer_hnaH · 2024-12-01
> > **Response to Authors' Rebuttal**
> >
> > Thank you for the response. It has addressed my concerns.

---

> > > ### Author Response · Authors · 2024-12-01
> > >
> > > Dear Reviewer hnaH,
> > >
> > > Thank you for your valuable review and positive feedback! We are pleased to hear that your concerns have been addressed. We kindly request you to consider raising your score, as the issues have now been resolved.
> > >
> > > Thank you,
> > >
> > > Authors.

---

### Official Review · Reviewer_7M3r · 2024-11-03

**Soundness:** 3
**Presentation:** 3
**Contribution:** 2
**Rating:** 6
**Confidence:** 4

**Summary:**

This paper proposes to enforce consistency of the sorting relationship between the student model and the EMA teacher model as a post-training strategy for advanced vision foundations models, such as DINOv2. The model is post-trained on the COCO or Pascal dataset. Improvements are observed in downstream semantic segmentation tasks on Pascal and COCO, especially under the evaluation protocol of linear probing.

**Strengths:**

1. The paper is clearly written and easy to follow.
2. The motivation of further improving advanced vision foundation models is promising, especially by designing a lightweight post-training technique.
3. The results under frozen encoders are impressive, compared with the original encoders.
4. The method of applying sorting-based consistency regularization is new to the post-training field.

**Weaknesses:**

1. My main concern lies in the difference between sorting-based regularization and a much simpler soft cross-entropy loss. It means, if we directly compute the similarity between $F_s \in R^{N \times d}$ and $F_r  \in R^{N_r \times d}$ as well as the similarity between $F_t  \in R^{N \times d}$ and $F_r  \in R^{N_r \times d}$ (suppose the obtained two similarity maps are $S_s, S_t \in R^{N\times N_r}$), can we directly apply a soft cross-entropy loss between $S_s$ and $S_t$ along the $N_r$ dimension? Will this be a stricter optimization target than the plain sorting-based strategy? Because it not only requires the sorted order to be the same, but also their pair-wise similarities with the referenced patches to be the same.

2. It is good to achieve promising results on downstream Pascal, COCO, and ADE20K datasets, when applying the post-training strategy. But I am wondering whether the post-training stage, especially only conducted on a small-scale COCO dataset, will potentially ruin the original DINOv2's good representation in broader scenarios, such as sketch images or urban-scene images (Cityscapes). It will be good to know how the post-trained model trades off between generalization and scene specification.

3. The proposed method fails to achieve obvious improvements over the original pre-trained encoders when evaluated under the fully-fine-tuning protocols (Table 4). For example, although the model is explicitly fine-tuned on COCO, it is still inferior to the original DINOv2 on COCO-Stuff dataset (46.8 *vs.* 46.5). Are there any insights into these results?

**Questions:**

- Could the authors provide additional results of semantic segmentation on urban-scene datasets like Cityscapes and remote sensing datasets?
- Could the authors explain why the fully-fine-tuning results are not better than the original pre-trained encoders? Indeed, it can be expected that the post-trained encoder is good than pre-trained encoder when evaluated under a frozen state, as the feature relationships are explicitly strengthened by the proposed strategy.

---

> ### Author Response · Authors · 2024-11-21
>
> **Applying cross-entropy on the similarity matrix instead of the sorted ones:**
>
> Thank you for highlighting this interesting potential ablation. We have now conducted an additional experiment where we remove the sorting algorithm, and simply apply the cross-entropy loss on the similarity matrices, as requested by the reviewer. The results are presented below:
>
>  *Table 1. Ablating the effect of sorting component. LS=Linear segmentation, IC=Vision in-context learning*
> | Method           | PVOC LS      | PVOC IC      | ADE20k LS       | ADE20k IC       |
> |-------------------|--------------|--------------|--------------|--------------|
> | Without sorting   | 47.3         | 17.8         | 15.8         | 5.1          |
> | With sorting      | **78.9**     | **62.0**     | **37.3**     | **21.7**     |
>
>
> The table is in the same format as our ablation studies (LS=Linear layer segmentation, IC=In-context segmentation). We find that removing the sorting component significantly reduces performance – though it does not lead to collapse. This can be attributed to the inability of a simple cross-entropy loss on similarities to provide sufficient discriminative power to separate dissimilar samples. Consequently, if the model is trained for enough epochs,  this may lead to solutions suffering from mode collapse, where all features increasingly become identical. While the EMA teacher helps mitigate this issue to some extent, our experiments indicate that it is insufficient on its own. Incorporating a sorting component that explicitly orders neighbors with discrete rankings leads to a more effective and stable solution.
>
>
> **Comparing the generalization on out-of-distribution datasets such as Cityspaces (Question 1)**
>
> As requested by the reviewer, we report the performance of various backbones, including DINOv2R fine-tuned with the NeCo loss, on out-of-distribution datasets such as Cityscapes (semantic segmentation) and NYUd (monocular depth estimation). The evaluation follows the protocol outlined in Table 2 of the paper.
>
>
>  *Table 2. Linear segmentation on Cityscapes (mIoU)*
>
> | Backbone    | Original | + NeCo   |
> |-------------|----------|----------|
> | Dino        | 40.7     | **43.7** |
> | iBot        | 43.0     | **44.8** |
> | TimeT       | 42.2     | **42.8** |
> | Leopart     | 44.5     | **45.5** |
> | Dinov2R     | 48.9     | **52.9** |
>
> ---
>
>  *Table 3. Linear depth prediction on NYUd (Root-mean-squared-error, lower is better)*
>
> #### **Backbone (ViT-S)**
>
> | Backbone    | RMSE     |
> |-------------|----------|
> | Dinov2      | 0.460    |
> | Dinov2R     | 0.456    |
> | Dinov2R+NeCo | **0.453** |
>
> #### **Backbone (ViT-B)**
>
> | Backbone    | RMSE     |
> |-------------|----------|
> | Dinov2      | 0.412    |
> | Dinov2R     | 0.410    |
> | Dinov2R+NeCo | **0.397** |
>
>
>  *Table 4. Linear segmentation on Vaihingen - Remote Sensing (mIoU)*
>
> #### **Backbone (ViT-B)**
> | Backbone          | Original | + NeCo |
> |--------------------|------|--------|
> | CLIP              | 25.7 | **28.8**   |
> | SigCLIP           | 26.9 | **28.3**   |
> | Dinov2R           | 34.2 | **35.7**   |
>
>
> As demonstrated in the results, NeCo consistently enhances the generalization of all backbones on the Cityscapes dataset. Furthermore, even for tasks that diverge from semantic segmentation, such as depth estimation, NeCo reduces the DINOv2R error by around 2%. These findings highlight that NeCo not only maintains but improves the generalization capability of DINOv2R features across diverse tasks. For completeness, we also include results from [1], showing that NeCo consistently enhances the performance, even on out-of-distribution datasets.
>
> **Why are the full fine-tuning results not significantly better than the original DINOv2R (Question 2)?**
>
> Actually, in 3 out of 4 full-fine-tuning experiments, including the COCO-Stuff dataset for the base variant, our model outperforms the base DINOv2R, as shown in Table 4. However, for the small variant on COCO-Stuff, indeed our performance is slightly lower. This might be attributed to the high potential of full fine-tuning to adjust all the model’s weights. The original DINOv2R checkpoint benefits from its pretraining on LVM142M with 142 million images, providing a significant advantage by capturing a more diverse range of visual patterns and features. In contrast, the NeCo training dataset contains only 118K images—just 0.11% of the DINOv2R training dataset—which explains the slightly lower performance (0.3%) in some experiments. We are sure that applying the NeCo loss on a varied dataset such as LVM142M would bring even larger benefits.
>
>
> We thank the reviewer once again for the valuable feedback. We hope the response adequately addresses the concerns raised and encourages the reviewer to improve the score.
>
> [1] - Markus Gerke, I. T. C. "Use of the stair vision library within the ISPRS 2D semantic labeling benchmark (Vaihingen)." Use of the stair vision library within the isprs 2d semantic labeling benchmark (vaihingen) (2014).

---

> > ### Comment · Reviewer_7M3r · 2024-11-28
> >
> > Thank the authors for the response. It has addressed my concerns. I will raise my rating to 6.

---

> > > ### Author Response · Authors · 2024-12-02
> > >
> > > Dear Reviewer 7M3r,
> > >
> > > Thank you for your valuable review and constructive feedback! We are glad to hear that our rebuttal addressed your concerns and appreciate your decision to increase the score.
> > >
> > > Sincerely,
> > >
> > > Authors

---

> ### Author Response · Authors · 2024-11-25
>
> Dear Reviewer 7M3r,
>
> Thank you again for the time and effort spent on your thorough review of our paper. Since the author-reviewer discussion deadline is fast approaching, we kindly ask for feedback on our responses. We would be happy to discuss more if there are still some open questions.
>
> Best Regards,
>
> Authors

---

### Author Response · Authors · 2024-11-27

Dear Reviewers,

We have dedicated considerable effort to conducting further experiments and meticulously addressing each of your comments. We are eager to hear your thoughts on our rebuttal and fully prepared to engage in further discussions regarding any additional concerns you may have.

Warm regards,
The Authors.

---

### Meta-Review · Area_Chair_dSyD · 2024-12-19

**Metareview:**

This paper introduces NeCo, a self-supervised training loss intended to enforce patch-level nearest neighbor consistency between student and teacher models. NeCo is a lightweight post-training technique that leverages differentiable sorting applied to pretrained representations to enhance the learning signal and boost overall performance. The results show significant improvements across various downstream tasks, achieving new state-of-the-art results in non-parametric in-context semantic segmentation and 3D understanding, all while requiring minimal training time.

After the rebuttal phase, the paper received consistently positive scores: 6, 6, 6, and 8, leading to an average score of 6.5. All reviewers praised the paper for being well-written, sound, and easy to understand, highlighting that the proposed NeCo is both simple and effective. Based on this feedback, the area chair recommends accepting the paper.

**Additional Comments On Reviewer Discussion:**

Reviews 7M3r, hnaH, and KF4P indicated that the authors' responses adequately addressed their concerns.

---

### Decision · Program_Chairs · 2025-01-22

Accept (Poster)